# Privacy risks of whole-slide image sharing in digital pathology

Petr Holub [1,2] ✉, Heimo Müller [3], Tomáš Bíl[2], Luca Pireddu [4], Markus Plass [3], Fabian Prasser [5], Irene Schlünder[6], Kurt Zatloukal [3], Rudolf Nenutil[7] & Tomáš Brázdil[8]

Access to large volumes of so-called whole-slide images—high-resolution scans of complete pathological slides—has become a cornerstone of the development of novel artificial intelligence methods in pathology for diagnostic use, education/training of pathologists, and research. Nevertheless, a methodology based on risk analysis for evaluating the privacy risks associated with sharing such imaging data and applying the principle "as open as possible and as closed as necessary" is still lacking. In this article, we develop a model for privacy risk analysis for whole-slide images which focuses primarily on identity disclosure attacks, as these are the most important from a regulatory perspective. We introduce a taxonomy of whole-slide images with respect to privacy risks and mathematical model for risk assessment and design . Based on this risk assessment model and the taxonomy, we conduct a series of experiments to demonstrate the risks using real-world imaging data. Finally, we develop guidelines for risk assessment and recommendations for low-risk sharing of whole-slide image data.

The last decade has seen tremendous advances in the methods available to pathologists for computer-assisted diagnosis, particularly thanks to the rapid developments in digital microscopy, which has reached high interchangeability levels with optical microscopy[1]; the whole field has become known as digital pathology[2]. The availability of large volumes of imaging and other types of clinically relevant data, as well as the availability of large-scale compute capacities has resulted in the massive development of Artificial Intelligence (AI) methods aiming to support pathologists in the diagnostic process[3].

The fundamental data used in this domain are whole-slide images (WSIs): high-resolution optical microscopy scans of the whole slide of biological material, resulting in image data typically in the order of gigapixels or even tens of gigapixels, as shown in Fig. 1. WSIs are widely used for purposes ranging from routine diagnostics to development and application of AI models. The images are commonly stored in databases linked to other types of the data—e.g., as a part of hospital

information systems—and they are sometimes shared under tight confidentiality agreements (e.g., ADOPT CRC-Cohort[4]) or as open data under the assumption of inherent anonymity (e.g., CAMELYON competition[5,6] or TCGA Digital Slide Archive (TCGA DSA https://cancer.digitalslidearchive.org/)[7]. WSIs from the same patients can also appear in different data sets associated with different data, and these can be potentially linked.

The process of creating a WSI begins with the acquisition of the biological material from a patient in a surgery or a biopsy. The material is then cut into blocks that are formalin-fixed and paraffin-embedded (hence the FFPE abbreviation), which are then sectioned and mounted onto glass slides and stained (colored) based on the type of the material and diagnostic methods to be applied—most common stainings being hematoxylin-eosin, van Gieson or various modern immunochemical staining methods. The material is then digitized using a slide scanner in the visible or fluorescence spectrum using a small pixel

[1]BBMRI-ERIC, Graz, Austria. [2]Institute of Computer Science, Masaryk University, Brno, Czech Republic. [3]BBMRI.at & Diagnostic & Research Center for Molecular BioMedicine, Medical University of Graz, Graz A-8010, Austria. [4]Visual and Data-intensive Computing Group, CRS4, Pula, Italy. [5]Berlin Institute of Health @ Charité – Universitätsmedizin Berlin, Berlin, Germany. [6]TMF eV, Berlin, Germany. [7]BBMRI.cz & Masaryk Memorial Cancer Institute, Brno, Czech Republic. [8]Faculty of Informatics, Masaryk University, Brno, Czech Republic. ✉e-mail: petr.holub@bbmri-eric.eu

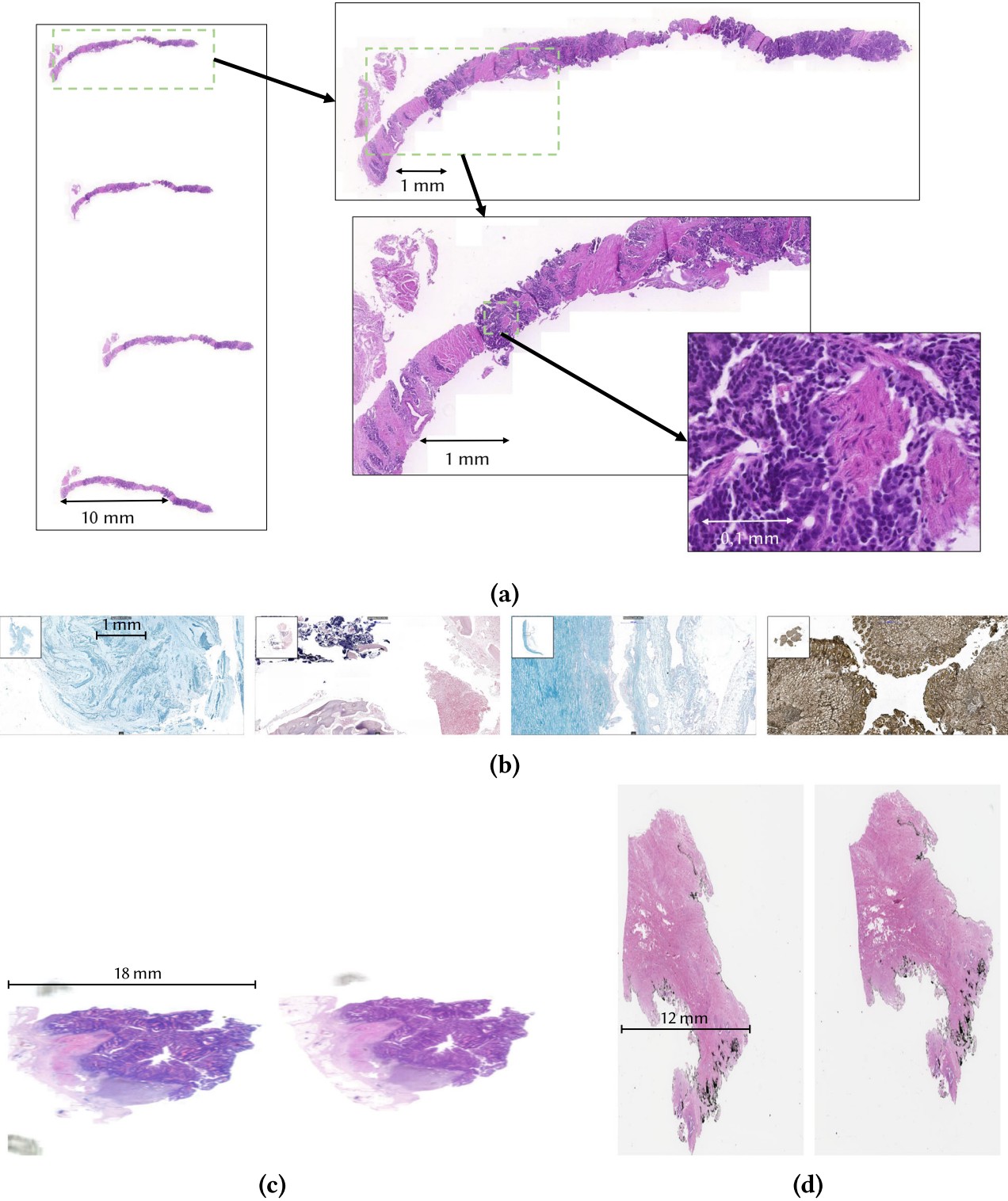

**Fig. 1 | Examples of WSIs and details at high magnification. (a)** Prostate cancer biopsy image has been stained with hematoxylin-eosin staining and scanned at 20×, resulting in resolution of approximately 100,000 px × 200,000 px. **(b)** Various types of less common staining methods, left to right: Giemsa, Gram, Alcian Blue Stain, and Warthin-Starry (silver) stain. All images have the same scale. **(c)** Example of the same colon tissue from MIDI and FLASH scanners (scanners specified in detail in Methods). Both images have the same scale. **(d)** Example of the consecutive slides. Both images have the same scale.

size such as 0.250 μm/px (generally designated as 20× magnification) or even 0.125 μm/px (usually denoted 40× magnification). The resulting images show the detailed cellular structure of the tissue, as illustrated in Fig. 1, and their high resolution results in an image size typically in the order of gigapixels or tens of gigapixels. In some cases, the scan can also include a visible patient identifier in the slide label—

e.g., a bar code that could be a patient-related ID or a pseudonym (a code of the patient used for a particular research purpose). Metadata in the image file(s) usually also includes details about the scanner and the settings used for the acquisition.

Given the large amounts of data required for the development of AI models, developing AI models for digital pathology requires access

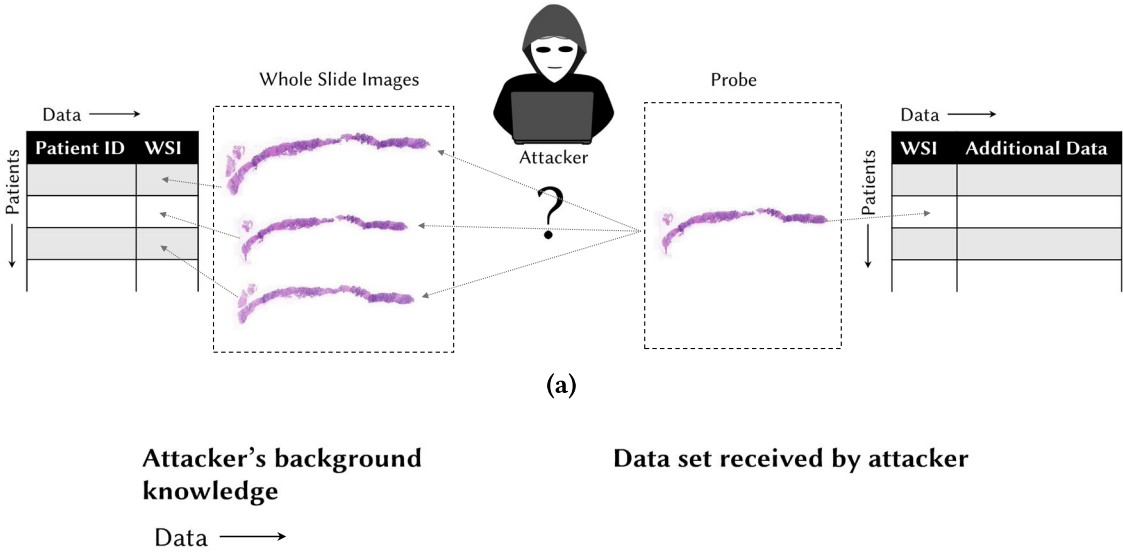

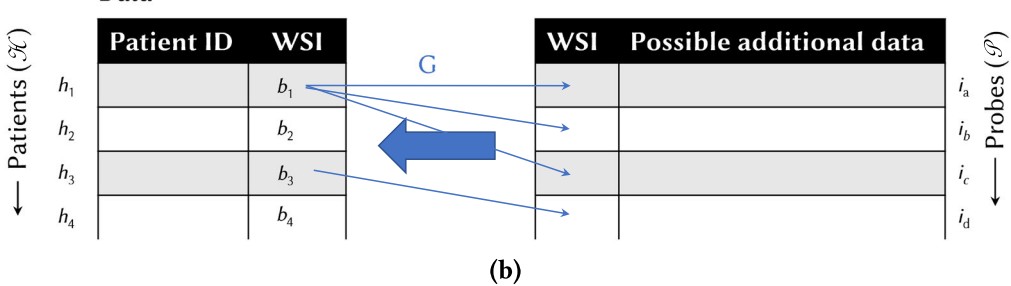

**Fig. 2 | Illustration of the assumed attack model. (a)** Assumed model for data structure—tabular data with patients in rows and different data types in columns, demonstrating that different data sets can be linked by a WSI. **(b)** Illustration of the probe ground truth function *G*, which defines which probes belong to which patients according to the "ground truth" (not known to the attacker): $G(h_1) = \{i_a, i_b, i_c\}$ and $G(h_2) = \{i_d\}$.

to large WSI collections, or even assembling collections by pooling data from different sources. However, the privacy risks related to WSI sharing have not yet been systematically explored and the practice of sharing is extremely heterogeneous: from the above-mentioned approaches considering WSIs low-risk data and sharing them as open data sets, to the opposite extreme considering them as sensitive as other clinical data and sharing them only as a part of pseudonymized data sets under contracts compliant with the applicable data protection laws, such as the General Data Protection Regulation (GDPR) in European countries.

Hence, the pivotal question is: what are the privacy risks related to sharing WSIs and are there any circumstances under which the risks can be considered low enough to treat the data as anonymous? At a first glance, it would appear unfeasible to identify the source individual from a WSI alone, which uniquely refers to the biological material that has already been removed from the patient's body and therefore histological slides and WSIs produced from this tissue cannot be directly reproduced from the patient. On the other hand, the WSIs are very large with relatively characteristic tissue structure, making each image very unique and potentially enabling the extraction of identifying information. There are also potential artifacts, such as slide preparation characteristics, associated data and metadata and various other factors that can significantly increase the likelihood with which a WSI can be traced back to its original FFPE slide and associated data.

## Results

In this section, we define WSI hierarchy and use it for evaluation of the worst-case probe attack success rate $R_s$ for the various WSI linking attacks. $R_s$ is a ratio of vulnerable patients to total number of patients, given that the attacker is using image similarity. Formal definition of

the attack model is the Methods section below. Results from linking attacks are used to formulate guidelines to releasing WSI data.

### Problem statement

We consider the following problem, illustrated in Fig. 2a. Assume that an attacker possesses a background knowledge consisting of a data set of patients indexed by a PatientID (for simplicity, we assume that PatientID is unique) and including WSI data linked to the patients. Now the attacker is given another WSI, called a probe, possibly associated with additional data. We assume that the probe belongs to one of the patients in the data set, but the probe itself does not necessarily appear in the data set (for instance, with respect to a WSI in the background, the probe may be a WSI of a different slice cut from the same block of material obtained from the patient). The aim of the attacker is to link the probe to the correct donor patient by matching it with a WSI from the same donor in the background knowledge.

Privacy risks are then given by the additional data associated with the probes; this additional data can be either derived from the WSIs or they can be merely associated with the WSIs. If the linking attack is successful, the attacker can link this additional data to the correct patient, thus expanding his knowledge about the individual. As a practical example of how this situation might emerge, consider a researcher who would like to train an AI model using digital pathology data to predict the prognosis of treatment result for melanocytic tumors with and without BAP1 mutation[8]. For this purpose, the researcher obtains a pseudonymized data set from Hospital X consisting of WSIs and the associated diagnosis, information about other cancers, patient outcome, and BAP1 mutation status. Suppose then that the researcher learns from a colleague at Hospital X that the institute is contributing data to a public archive, such as TCGA DSA, where they publish data sets consisting of WSIs and associated rich

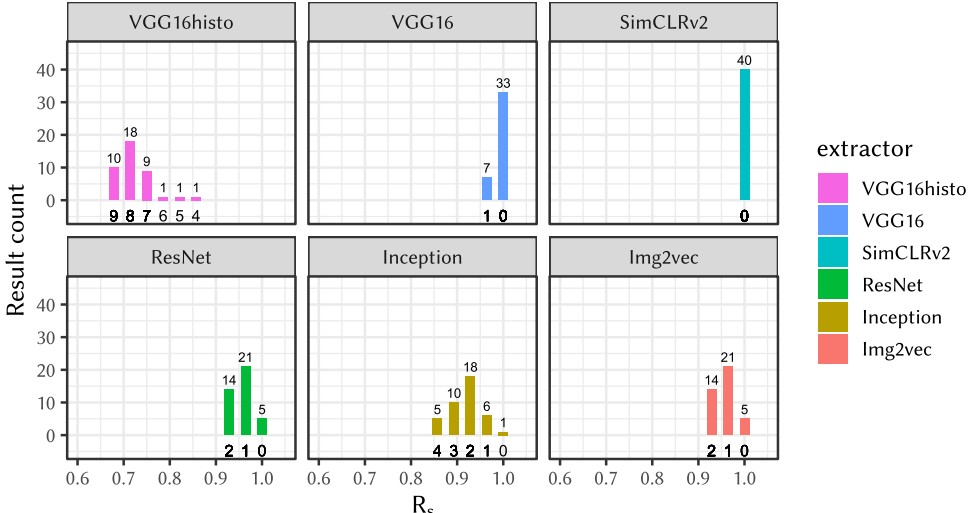

**Fig. 3 | $R_s$ on tissue slides scanned on different scanners.** In this experiment the $R_s$ has a low number of distinct values and hence we are providing full results visualized as a bar plot with the number with counts of misclassified results. The $R_s$ is on the horizontal axis; the bold numbers under the bars indicate number of misclassified patients (i.e., 1 - number of $f$-vulnerable patients); the vertical axis and the small number above the bar indicates the number of occurrences out of total $Y = 40$ independent experiemnts on $X = 28$ slides as described in (E-1) and Statistics and Reproducibility.

genetic data describing at least the status of several different mutations. Now, suppose that the researcher's data set and the contributions to the public archive's collection contain data from the same patients—regardless of whether the same WSIs are in both sets. The researcher might then be able to enrich his knowledge about the patients in the data set he received from Hospital X by matching the WSIs with the public data set and linking rich genetic data to the clinical data he received. Depending on the extent of the information in researcher's background, it might be possible to make a good guess about whether the patient has been included in the public data set. The researcher knows that: the patients in the AI data set have consented for research, they fall within recent year range when the mutations were already tested, the number of patients with this cancer and mutation status is low. This information as well as the BAP1 status can be used to facilitate an attack by more effectively targeting patients that the attacker is reasonably sure to find in the public data set. Moreover, this type of scenario is an example where stratification of patients in personalized medicine leads to creating very small populations—so small that even specialized regional cancer centers can have as few as 5 cases in a year and research is done on individual cases[9]. Therefore, the risk of having the researcher access more data than authorized and approved can become a plausible scenario.

The privacy risks associated with sharing this type of data can be seen as composed of two orthogonal dimensions: the likelihood of a successful attack and the harm resulting from a successful attack. As stated in the problem statement in the previous paragraph, this work focuses on the analysis of the likelihood dimension for linking disclosure attacks, where the WSIs can act as a key for linking different data sets—i.e., linking the WSIs in background knowledge to the set of WSI probes, possibly associated with additional data. On the other hand, the harm dimension represents a generic problem and depends on the type and total amount of information that the attacker is able to link to the same PatientID, drawing from both probes and their associated data as well as the attacker's background knowledge. While the harm aspect is not part of our attack model and experimental evaluation, it is still considered in the guidelines developed in the Discussion section of this paper.

In this work, we consider the following crucial questions. (1) Which WSI can be used by the attacker as a potentially effective probe? We introduce a hierarchical taxonomy capturing "closeness" of different WSIs with respect to their linking potential, starting with

identical WSIs, progressing through different scans of the same slide and WSIs coming from tissues with different degree of spatial or temporal closeness. (2) How to quantitatively measure likelihood of an attacker's success? We introduce a metric, which measures the probability of successful identification for deterministic attacks and which can be easily extended to randomized attacks.

Based on the metric and the slide hierarchy, we perform a series of experimental evaluations using state-of-the-art image similarity techniques, to demonstrate that the attacker's likelihood of success depends on the slide's location in the taxonomy. The experimental evaluation utilizes the extensive WSI collections of the Medical University Graz and the Masaryk Memorial Cancer Institute to evaluate likelihood of a successful attack in real-world settings. We outline guidelines for the conscientious sharing of WSI data, encouraging the advancement of data-driven research while protecting the identity of the individuals from whom the tissue was originally obtained. Our risk assessment methodology also provides practical approaches for data controllers to perform related analyses for their own data sets.

## WSI hierarchy

To analyze WSI linking attacks, we need to examine the factors in the image generation process that can make two WSIs—i.e., one in the attacker's background knowledge and one a probe—more or less easily linked to each other. The first factor is when and from where the tissue on the slide was extracted. We introduce the following *spatiotemporal hierarchy* of cases, ordered by decreasing potential image similarity:

(1) WSIs of the same slide *scanned with different parameters* (including age of the slide at time of scanning, scanning parameters such as resolution, type of scanner, etc.; see Fig. 1c);

(2) WSIs from the *same tissue block—which* can be further divided into consecutive (adjacent, see Fig. 1d) vs. distant slides;

(3) WSIs from the *same primary sample*, defined in ISO/DIS 20658(en), Definition 3.17, as "a discrete portion of material, intended for examination, study or analysis of one or more quantities or properties. It is retrieved during an acquisition procedure such as a surgery or a biopsy."; the same primary sample can be divided into multiple different blocks;

(4) WSIs from *different primary samples* (such as the primary tumor site, the lymph node, the metastasis) from the same patient *taken at the same time*;

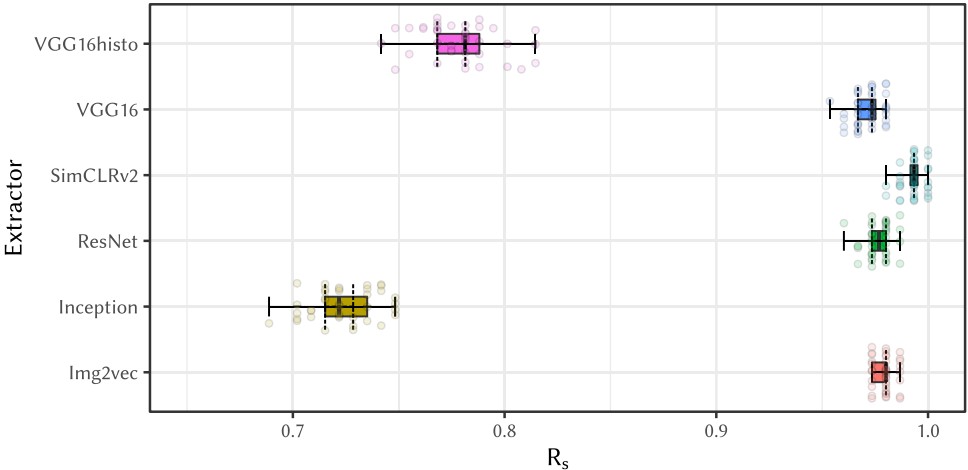

**Fig. 4 | $R_s$ on consecutive slides, with different feature extraction models and cosine distance−as measured in (E-2a).** Except for when using the Inception model as a feature extractor, the attacks were almost always able to successfully link probe slides to the correct donor patient. Done on X = 151 slide pairs in Y = 40 independent experiments. Boxplot settings is described in Statistics and Reproducibility.

(5) WSIs from different primary samples from the same patient *taken at different times*, which can also relate different diagnoses of the same patient.

The second factor in the image generation process affecting image similarity is the staining applied in slide preparation. We introduce the following hierarchy of staining cases, ordered by decreasing potential image similarity, which can be combined with the spatio-temporal hierarchy to reason about risks:
(1) (H-a) WSI slides from the same staining batch;
(2) (H-b) WSIs from different staining batches using the same staining method;
(3) (H-c) WSIs from slides stained using different staining methods (e.g., H&E vs. DAB-based immunohistochemistry vs. van Gieson).

This hierarchy led to the design of experiments further denoted (E-1) for the same slides scanned on different scanners corresponding to (H-1), (E-2a) for consecutive slides and (E-2b) non-consecutive slides from the same block (H-2). Details of the expreriment setup is in the Methods section below. Hierarchy elements (H-3), (H-4), and (H-5) were not experimentally evaluated as the methods tested already showed a dramatic decrease in $R_s$ even in (H-2), with increasing spatial distance of slides from the same primary sample. Peformed experiments included both linking attacks on full WSIs where the linking can use also the overall shape of the tissue on the slide, and crops that are only showing internal part of the tissue.

## Results on WSI linking attacks
First, we present results of linking attacks on entire WSIs. Fig. 3 presents the histogram of $R_s$ values measured in experiment (E-1), which measures the effect of the different slide scanning conditions on $R_s$. The results show that WSIs of the same tissue acquired using different scanners are very similar and can be used in a successful WSI linking attack.

Figure 4 presents the box plots of the $R_s$ values measured in experiment (E-2a), which tests WSI linking attacks using WSIs of consecutive slides scanned under the same conditions. The high feature similarity of consecutive slides also leads to successful WSI linking attacks in this case−if using the right neural network as a feature extractor. So, in this case as in (E-1), different WSIs can be easily linked to violate privacy of the patient.

Finally, Fig. 5 presents the box plots of the $R_s$ values measured in experiment (E-2b), which tests WSI linking attacks using WSIs of non-consecutive slides as shown in Fig. 1d. We have measured the effect of the spatial distance between the probe WSIs and the patient's "pivot" WSI on the $R_s$ value by varying the minimum distance threshold $l$ from 3 mm to 18 mm. Note that the larger the distance between cuts, the fewer WSIs are available as probes; hence the plot also shows the average number of probes per patient. As expected, the attack success value $R_s$ decreases with the increasing physical distance of cuts, as can be seen from drop in $R_s$ shown in the figure.

As the efficiency of our similarity-based linking method apparently depends on the number of patients and the number of probes for each patient, we have evaluated the effect of these parameters on the $R_s$ value in the case of non-consecutive slides−i.e., in experiment (E-2b).

For evaluating the dependency on number of patients, we select a random subset of $n$ patients and, for each patient, we randomly select one slide and insert it into $\mathcal{B}$, while we insert the rest of their slides in $\mathcal{P}$; we then measure the corresponding $R_s$ value. We performed this process 40 times for every number of patients $n$ between 1 and 80. The resulting $R_s$ statistics are presented in Fig. 6a. Given the prosecutor attack model, the $R_s \approx 1$ for small number of patients (<5) and drops to still significant $R_s \approx 0.5$ for the maximum of 80 patients in our data set.

For evaluating the dependency on the number of probes per patient, we tested attacks varying the number of probes per patient from 1 to 6. For this experiment, from our data set we selected the set of patients with at least 7 slides, leaving us with 43 patients (hence the maximum of 6 probes per patient tested, as increasing this upper limit further would dramatically decrease number of patients available for the experiment). For a given number of probes to be tested $p$, for each patient we randomly selected a subset of $p+1$ slides; of these, a random slide is placed in $\mathcal{B}$, while the rest are placed in $\mathcal{P}$. We then proceed with the measurement of $R_s$. The experiment was repeated 40 times for each number of probes $p$. The resulting $R_s$ statistics are shown in Fig. 6b. One can observe that $R_s$ increases substantially with the number of available probe slides $p$. This effect is explained by the random slide selection process−increasing $p$ increases the probability of including a "strong" probe slide that falls in the hierarchy case (H-2) with low distance, for which the attack implementation has shown to be most effective (see Fig. 5). This effect implies an increasing residual risk as more WSIs from the same block of tissue are released, even if they are sampled far from each other.

In order to examine linking attacks when overall shape of a WSI cannot contribute to extracted features and hence to the similarity, we present results for linking WSIs cropped to an internal tissue only as

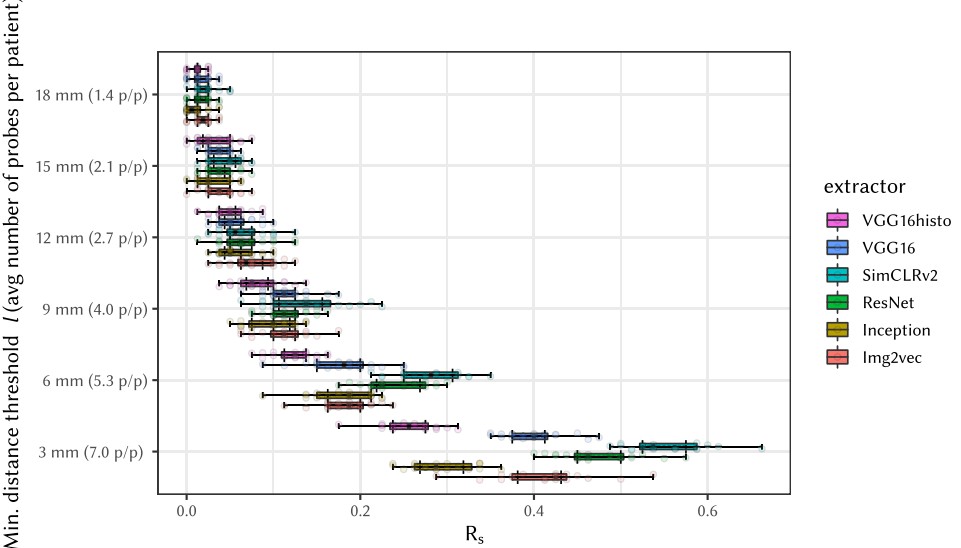

**Fig. 5 | Effect of distance between probe and target slides on $R_s$ measured in experiment (E-2b) on attacks using non-consecutive slides.** The labels on the vertical axis denote the minimum distance threshold $l$ (in mm) and, in parentheses, the average number of probes available per patient (p/p), which decreases with increasing threshold $l$. Set of total of X = 558 slides for 80 patients (subset selected based on distance tested) was used in Y = 20 independent experiments. Boxplot settings is described in Statistics and Reproducibility.

shown in Fig. 7. Figure 8 summarizes our results on cropped WSI linking attacks measured on consecutive slides in experiment (cE-2a). $R_s$ decreases only slightly when compared to the results of experiment (E-2a) in Fig. 4 (i.e., WSIs of consecutive slides scanned under the same conditions). When comparing results of Inception, we can see that cropping tissue actually significantly increases $R_s$; we assume this is caused by the changes in shape/border of the tissue on consecutive slides, to which this feature extractor seem to be more sensitive compared to other extractors.

Similarly, we have also tested attacks on cropped WSIs of non-consecutive slides in experiment (cE-2b); results are summarized in Fig. 9. Compared with results of experiment (E-2b) in Fig. 5 on whole non-consecutive images, there is again a slight decrease in $R_s$. Crop shifts have not been evaluated for non-consecutive slides, as the slide distance on cropped slides already drives $R_s$ down to minimal levels and shifting has marginal room for effect.

We can summarize that the success of a (full) WSI linking attack using the presented methods does not depend significantly on the availability of the border and on the overall shape of the tissue, but the internal structure of the tissue is sufficient. The used attack model implementation is also relatively insensitive to small shifts in the crop area. Only after shifting by 50 pixels (area overlap 71% to 78% depending on the shift direction) or more, the $R_s$ value deteriorates below 0.8 for all extractors except SimCLRv2, where the significant drop starts only at 75 px (area overlap 58% to 67%). This is expected behavior as SimCLR/SimCLRv2 are trained using random image cropping as a part of stochastic data augmentation.

## Discussion

WSI are a specific category of data from the privacy risk perspective. In this paper, we have primarily focused on linkage attacks, which may lead to unintentional identity disclosure or enable inferring additional information about the data subjects. As demonstrated by our experimental results, with a relatively straightforward attack model, the potential for identical WSIs to act as links across different data sets is substantial; likewise for WSIs generated from the same or from closely spatially related physical slides. Additional insight into risks could be obtained by organizing an international competition, where different teams

could propose and compare potentially stronger attack methods. The authors propose to organize such a competition using the attack model presented in this paper to develop and test proper technical safeguards for protecting WSI data donor privacy.

Moreover, an aspect which has not been a specific focus of this paper, apart from generic feature extraction, is how much information can be inferred directly from WSI data—i.e., what additional information can be inferred from the "fingerprint" itself. Many AI models have been developed to infer diagnoses (see survey by Pocevičiūtė et al.[10]) and disease-specific scores such as a UICC stage for colorectal cancer or a Gleason score for prostate cancer[11,12]. But AI models have been shown to also recognize less apparent features, such as information on particular mutations[13–16] or even yet unknown morphological features of prognostic significance[11]. Inferring such information might be used to further improve linkage attacks (e.g., if mutation information is known to the attacker) or to infer additional information about the data subjects, such as clinical or genomic data. When releasing WSI data as a part of bigger data sets, the likelihood of inferring sensitive data should be taken into account; this aspect is considered in the proposed guidelines presented in the following Section. This is relevant primarily when releasing the data as de facto anonymous data sets, where it can be correlated with other data sets in the future (e.g., genomic data).

To summarize, given the experimental findings and the additional considerations presented in this text, when WSI data is to be released across different public data sets—i.e., without controlled access and additional contractual responsibilities—caution needs to be exercised to mitigate privacy risks. At the same time, we acknowledge the need to maximize openness of data and to make large volumes of data available for the development of AI models and other research that have the potential to significantly improve health care. Hence, we propose the risk assessment models and data release guidelines for WSIs presented in the following Section.

When releasing data sets containing WSIs, it is desired to publish them as openly as possible to support their reuse, but also to keep the data as closed as necessary. In the following subsections we define the relevant terminology, discuss risk assessment aspects, and propose two sets of guidelines: the first for releasing WSI as de facto anonymous data sets, and the second for releasing pseudonymized (or even

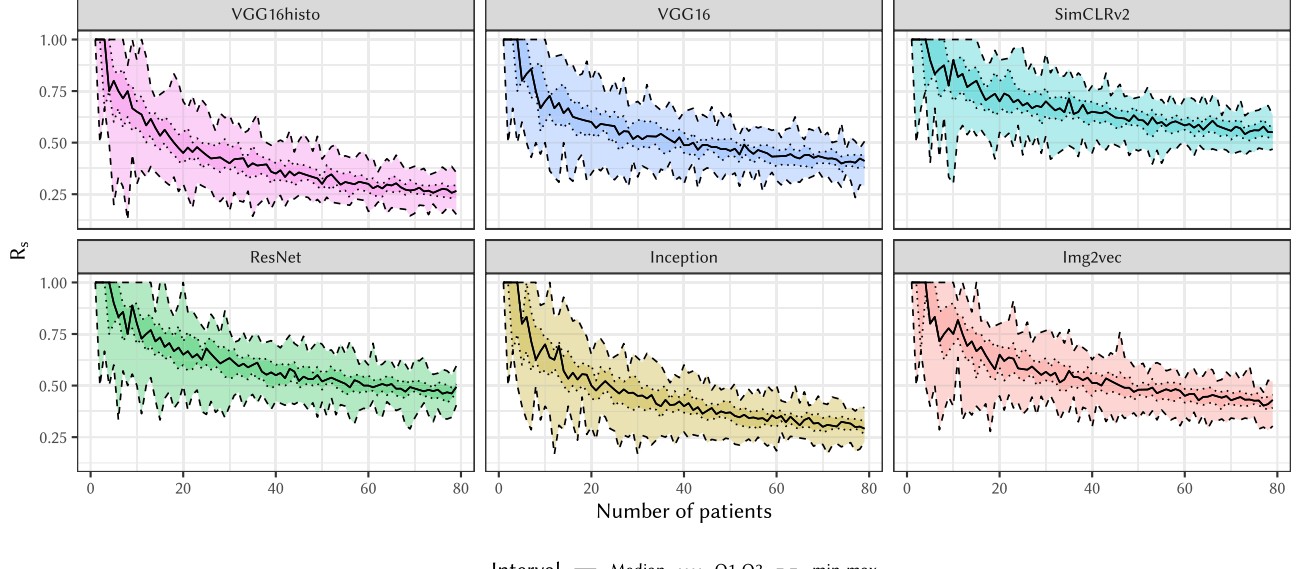

(a)

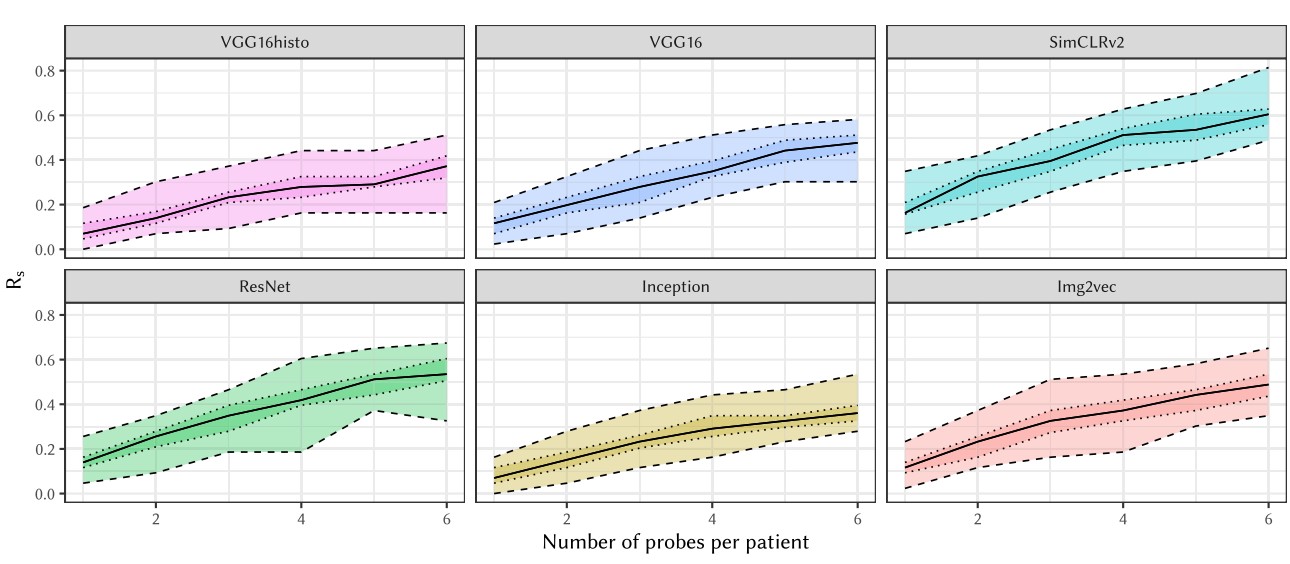

(b)

**Fig. 6 | Effect of the number of patients and number of slides (probes) on $R_s$.** Figure (**a**) shows the $R_s$ measured in attacks on data sets including data from 1 to 80 patients. On the other hand, Figure (**b**) shows the observed growth in $R_s$ as the number of available probes increases—from 1 to 6 in these experiments. These two experiments use non-consecutive slides (H-2) described in (E-2b). Set of total of X = 558 slides for 80 patients (subset selected based on number of patients or slides tested) was used in Y = 40 independent experiments. Note that, as in the other figures in this article, these modified box plots show median, quartiles, and min/max.

identified) data set under a contract defining appropriate technical and organizational safeguards. Note that the privacy risk analysis in this paper is independent of any particular jurisdiction. We do, however, anchor the guidelines developed below to the terminology established by the GDPR.

The notion of de facto anonymity captures the idea that it is reasonably unlikely that the person subject of the data could be identified. We can examine the notion in more detail by reading the GDPR. In recital 26 it defines anonymous information as:

…information which does not relate to an identified or identifiable natural person or to personal data rendered anonymous in such a manner that the data subject is not or no longer identifiable.

To understand this definition, we need to understand what constitutes *personal data* and what makes a person identifiable. The GDPR defines personal data in Art. 4 (1) as:

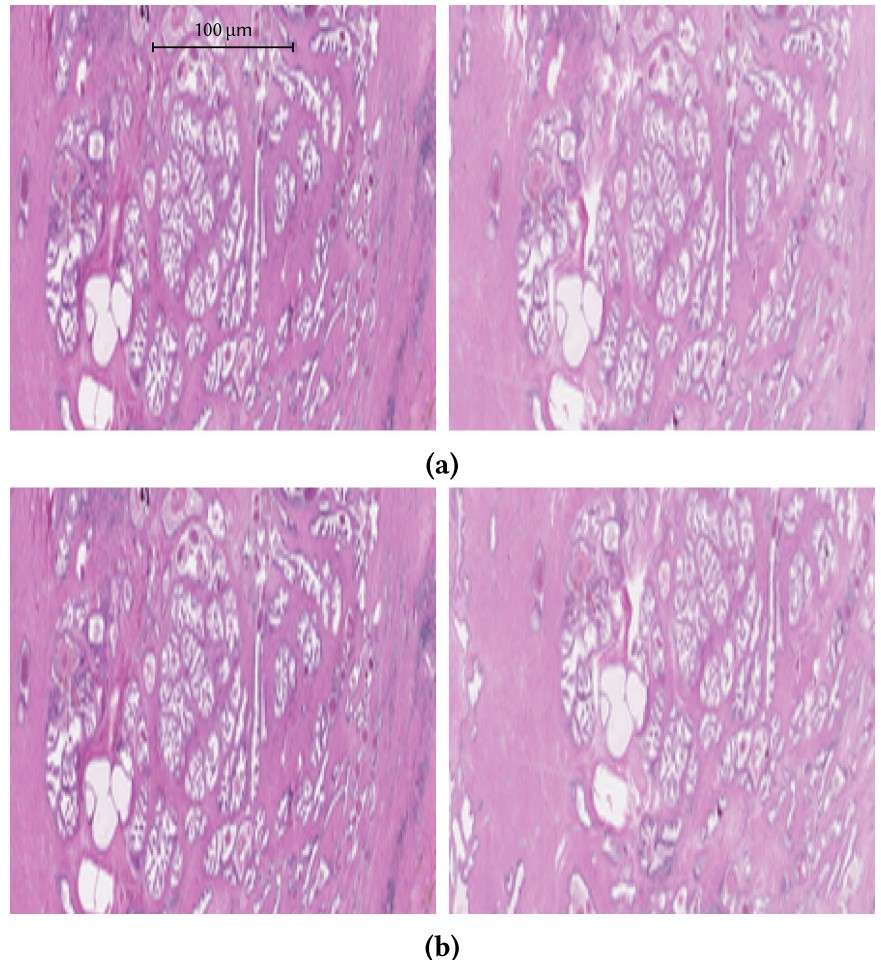

**Fig. 7 | Examples of consecutive slides used for cropped WSI linking attacks without shift and with a shift.** Figure **(a)** demonstrates cropped WSI without shift, **(b)** shows a shift of 50 px. The overlapping area of the images with a 50 px shift is 71% to 78%, depending on the shift direction. Note that the images shown are already downscaled to fit the input of the network, as described in Design of Experimental Evaluation of WSIs Linking Risks. Both images have the same scale.

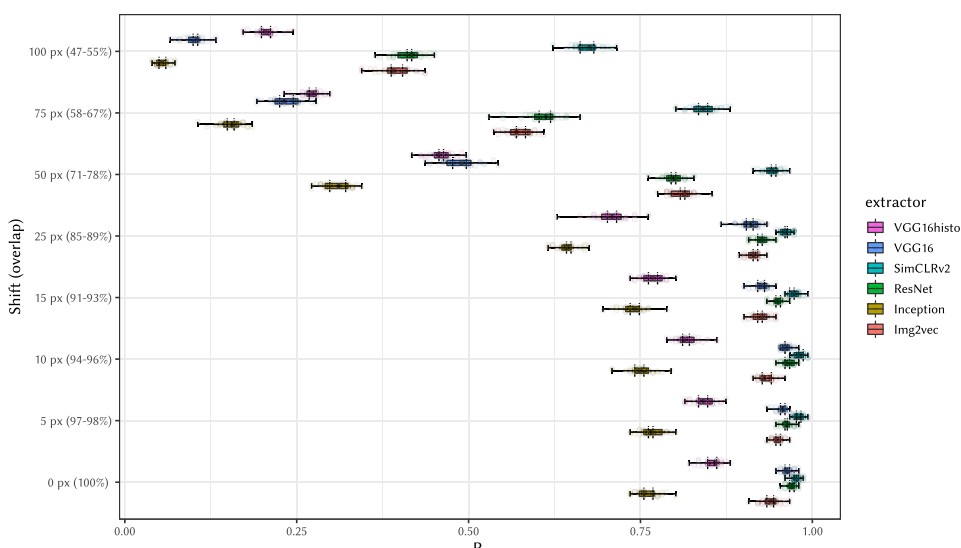

**Fig. 8 | Effect of the crop area shift on $R_s$ for cropped WSI of consecutive slides.** We observe that attacks are relatively insensitive to small shifts in the crop area; only after shifting by 50 px (area overlap 71% to 78% depending on the shift direction) or more, the $R_s$ value deteriorates significantly. Set of total of X = 151 slide pairs was used in Y = 40 independent experiments. Statistical evaluation and box-plot settings is described in Statistics and Reproducibility.

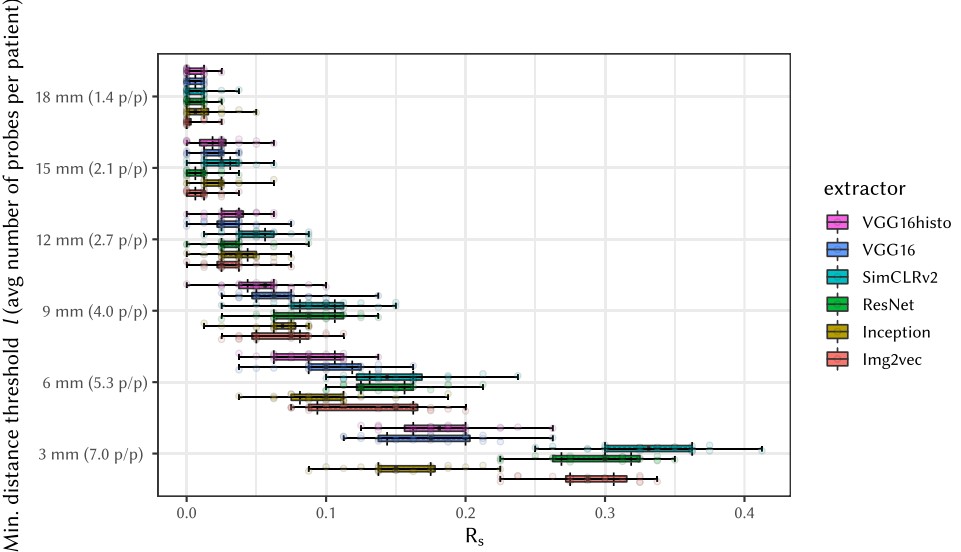

**Fig. 9 | Effect of distance between cropped, non-consecutive probe and target slides on $R_s$, measured in experiment (cE-2b).** The labels on the vertical axis denote the minimum distance threshold $l$ (in mm) and, in parentheses, the average number of probes available per patient (p/p), which decreases with increasing threshold $l$. Set of total of X = 558 slides for 80 patients (subset selected based on number of patients or slides tested) was used in Y = 20 independent experiments. Boxplot settings is described in Statistics and Reproducibility.

"'personal data' means any information relating to an identified or identifiable natural person ('data subject'); an identifiable natural person is one who can be identified, directly or indirectly, in particular by reference to an identifier such as a name, an identification number, location data, an online identifier or to one or more factors specific to the physical, physiological, genetic, mental, economic, cultural or social identity of that natural person".

Finally, recital 26 of the GDPR also provides important information on how to decide whether a natural person is identifiable:

"To determine whether a natural person is identifiable, account should be taken of all the means reasonably likely to be used... To ascertain whether means are reasonably likely to be used to identify the natural person, account should be taken of all objective factors, such as the costs and time required for identification, taking into consideration the available technology at the time of the processing and technological developments."

Further, according to the Art. 29 Working Party (predecessor of European Data Protection Board under the GDPR) even organisational measures can influence the status of anonymity (Art. 29 WP opinion 136, concept of personal data, p. 17), since identifiability depends on the background knowledge of potential attackers: the same data might be anonymous in one setting and personal data in another. Examples of such organizational measures include: access control and contractual obligation to not re-identify research participants, to not share the data with third parties, and to make them internally accessible only under confidentiality obligations, provided that the contractual party is reliable and able to fulfill those obligations.

To summarize, the stance taken by the GDPR is that anonymity is not an absolute value; it follows that absolute anonymity once and forever with zero risk of re-identification is not required by the GDPR. Instead, certain residual risks are acceptable and can be calibrated against the sensitivity of the data in respect to the impact of privacy breaches for the data subjects. This is captured by the notion of de facto anonymity.

Based on the risks demonstrated in the previous sections, we define recommendations for releasing WSI data as de facto anonymous data or as pseudonymized data, based on the concepts of data protection in European GDPR framework.

**Guidelines for releasing WSI as de facto anonymous data**
The following technical and organizational measures shall be considered before releasing WSI data in de facto anonymous data sets.

A1. *Minimize metadata.* Each WSI shall be stripped of any patient-identifying metadata. Technical metadata related to scanning processes should be reviewed and minimized for the given purpose (e.g., removing location identifiers if present, but retaining information on scanning parameters).

Rationale: Patient-identifying metadata, such as health insurance identifiers, would be a direct source of a patient's identity. Similarly, hospitals often use unique identifiers for the histopathological process, which is a unique patient identifier with one or more levels of indirection. Many relatively identifying technical metadata, such as serial number of the WSI scanner and location metadata, are typically not strictly needed or used for research activities with WSIs. This follows directly from the nature of the metadata and needed not be considered in the experiments presented in this work. Note that removal of patient-identifying metadata implies breaking provenance chain and hence traceability of data, as discussed in the provenance information management paragraph below. It also disables handling incidental findings.

A2. *Dissociate from patient records.* Both the WSIs to be released in a de facto anonymous data set and their originating slides must be dissociated from the patient records (except for the data which is released in the same de facto anonymous data set and is also subject to anonymization).

How the dissociation is done depends on the context of the de facto anonymization. It can be done using a legally enforceable contract preventing the recipient of the data from accessing the link. If this is not possible, the dissociation has to be done by removing all references between the WSIs/originating slides and any patient records in the data holder's/controller's information systems.

Rationale: Experiment (E-1) demonstrated that WSIs generated from identical physical slides on different scanners (i.e., (H-1)) can be

trivially linked. Hence, not only is the dissociation of WSIs necessary, but the originating slides must also be dissociated, so that it is no longer possible to release additional patient data that would be associated with that anonymized slide or its WSIs.

A.3. *Consider combinations with other data sets*. (a) If a WSI is released into more than one de facto anonymous data set, all corresponding records (i.e., WSI and data linked to it) in these data sets shall be considered linked. Rationale: This follows from ability to do bit-by-bit match of identical WSIs or more effectively by comparing hashes of WSIs as discussed in Characterization of privacy risks considered. (b) If the same WSI is released into a de facto anonymous data set and other non-open data sets, gaining information from the de facto anonymous data set is technically trivial for the recipients of the non-open data set and must be considered from the risk assessment perspective. Rationale: This follows from ability to do bit-by-bit match of identical WSIs or more effectively by comparing hashes of WSIs as discussed in Characterization of privacy risks considered.

Note that this may be a complex task when data releases can be done by different organizations—e.g., when a chain of data controllers is set up, in which several controllers are allowed to further release the data in whole or even just in part.

A4. *Slides from the same block*. If the WSI is a part of a series of consecutive slides, the same rules A1–A3 on dissociation apply to the whole consecutive WSIs stack and corresponding original slides.

Rationale: Experiment (E-2a) linking consecutive slides using common feature extractors showed high risks of linkage; even crops from the consecutive slides in experiment (cE-2a) were demonstrated to provide very good linkage if the overlap of areas is higher than 70%. For non-consecutive slides in experiment (E-2b) and their crops (cE-2b) the linking capability is much lower and decreases rapidly with the distance between slides in the stack (Fig. 5 and 9), with increasing number of patients, (Fig. 6a) and with decreasing number of slides per patient (Fig. 6b), though we can realistically anticipate improvements of the matching methods in the future. When only releasing distant non-consecutive slides, precaution needs to be taken for future data releases, so that intermediate slides are not released in other data sets. In the extreme case, this would result in consecutive slides being found in different data sets and matched easily. As a further consequence, FFPE tissue blocks from which multiple slides can be generated also have to considered as a means for generating linking information.

Note: If effective methods for linking more distant elements in the slide similarity hierarchies are developed in the future—i.e., for (H-3) to (H-5) or for different staining relationships (H-a) to (H-c)—this recommendation will be affected and will need to be expanded to cover new risks.

A5. *Consider information inference risks*. The probability of successfully inferring information from the WSIs shall be considered and in case that such information could practically lead to singling out a patient, the WSIs shall not be released as anonymous material. Examples include rare cancer diagnoses and rare mutations, where inferring these and their combinations from a WSI might narrow down the number of possible donor patients dramatically and even lead to singling out individuals. Qualified risk assessments need to be carried out according to state-of-the-art methodologies (e.g., those by Ohmann, et al.[17] or by El Emam[18]).

Rationale: As discussed above, there is increasing body of work on deriving information from WSIs. The state-of-the-art needs to be monitored and information derived with high reliability should be considered as if it is accompanying the WSI. Hence, a risk assessment needs to be done for the compound of the WSI and the derivable information.

A6. *Consider small populations*. When releasing slides from small (sub)populations, such as in case of rare diseases or highly stratified major diseases, for which inclusion criteria of the cohort may already indicate very low number of patients included, the risk assessment must consider this aspect. A rare disease diagnosis itself, for example, may already act as a partial identifier. A decision must be taken as to whether releasing the data as a de facto anonymous data set is acceptable, considering that an attacker may gain knowledge about members of the population from other sources, and what harm can be caused by deriving information from the de facto anonymous data set.

Rationale: Results in Fig. 6a show that with the prosecutor attack model, when the attacker knows the patient is in the data, the success rate increases rapidly for small populations. The prosecutor model implements the worst-case scenario for small populations, where the attacker gains membership information elsewhere and has it in his background knowledge.

A7. *Contract setup*. When releasing data as de facto anonymous, a legally enforceable contract shall be in place between the data controller and the data recipient, which prohibits re-identifying data subjects and/or inferring data about any specific individuals. This is a legal measure to mitigate residual risks, such as risks arising from the development of novel re-identification techniques for WSI data. Such legal measures may include limiting the purpose of the data processing as well as setting time constraints for the processing in order to manage risks related to the development of new methods in the future. It is recommended that the contract anticipates reproducing research results: instead of requiring the deletion of the data by the recipient after completing the research, it should allow the data to be kept for archival and reproducibility analysis/testing purposes.

When releasing WSI data in an open data set (e.g., publicly downloadable on the internet), at least a lightweight contract, such as a licence agreement for the data set, shall be implemented to which the recipient of the data set must actively agree.

Note to A5 and A6: These guidelines should not be interpreted as preventing sharing of WSIs from rare disease patients, but only relates to precautions to be taken for sharing data as de facto anonymous. The rare disease patients are known to be highly interested in the medical research that could help them and fellow patients and are generally positive to FAIR data sharing or even open data sharing[19]. Such data should be shared as personal data with the necessary awareness about the risks and adequate technical and organizational measures (see below). For other vulnerable or low-incidence patient populations, the attitudes might be less positive to data sharing and, hence, sharing the data as personal data with adequate measures in place is even more important in order not to compromise the trust placed by them on the researchers.

### Guidelines for releasing WSIs as personal data

When there is a legal basis for processing personal data for a particular purpose, it is recommended to release the WSIs and any other necessary linked data as personal data that is not anonymized. Of note, releasing data as personal data (even if pseudonymized) has the advantage of enabling research results to be fed back to the donors—for instance in the form of acting on incidental findings to prevent the development of a disease not yet known to the donor. The following technical and organizational measures shall be considered before releasing WSI data as personal data.

1. Regulate data transfer and/or processing with a legal contract. Use a legally enforceable contract to regulate the conditions under which access to the personal data is granted. Under the GDPR, this can be a data transfer agreement (i.e., establishing controller to controller data transfer) or a data processing agreement, regulating the data processing activities that are to be performed on behalf of a data controller.
2. Pseudonymize and minimize the data set. Direct patient identifiers should be replaced by pseudonyms. The data set should be minimized for the purpose for which it is being shared.
3. Safeguard data via technical and organizational measures. Processing of personal data is typically safeguarded by a combination

of technical measures (e.g., encryption of transmission channels and storage, network protection mechanisms, mechanisms for data destruction after termination of the contract) and organizational measures (e.g., organizational life cycle of the data, restriction of access to defined personnel and having appropriate contracts setup with each person authorized to access the data), which are defined in a contract between the provider and recipient(s) of the data. The contracts also need to restrict the purpose of use and define requirements data processing.

4. Consider information leakage from AI models. When AI models are developed using WSI data, it needs to be ensured that the model does not leak personal information learned from training data (see, for instance, the work by Shokri et al.[20]). Methods like PATE[21] can be used to mitigate these risks.

Provenance information is an important aspect of trustworthy data with defined and analyzable quality[22]. For personal/pseudonymous data sets, provenance information can lead back to the originating biological material, but the link to the donating patient can typically only be resolved by authorized personnel at the source organization. When releasing de facto anonymous data sets, provenance information can only start with the anonymization process and cannot link to the originating data (this link must be intentionally and permanently destroyed).

Please note that our guidelines only consider aspects specific to WSI data, and they need to be complemented by relevant common best practices for processing personal data. Under GDPR this entails that there has to be a legal basis for processing personal data, such as consent or performance of a contract (list of possible legal bases is specified in Art. 6 of GDPR) and the data subject needs to be able to exercise their rights (e.g., right to be informed, rights of access, rectification, erasure, restricting processing). Moreover, though anonymized data is not personal and processing it does not require a legal basis, the anonymization process itself−i.e., generation of anonymized data from personal data−is just a specific form of personal data processing and, if done, there needs to be a legal basis for it, too. The details of the process are dependent on the relevant jurisdiction and are outside of the scope this article.

## Methods
### Characterization of privacy risks considered
In our model we consider privacy risks caused by WSIs acting as accurate or approximate links across data sets. The attack model assumes that an attacker that has a background knowledge, including WSI data, receives a WSI probe with associated additional data. The attacker tries to correctly assign this probe to the patients in the background knowledge. This operation can be performed deterministically−e.g., based on image similarity. Note that the deterministic assignment is a special case of the randomized one where each probe is assigned to a single patient with probability one. We demonstrate how the deterministic model can be extended to a randomized one, but this is not necessary for the experimental evaluation used in the paper.

**The intuition behind the metric.** We define the attack success rate to be the proportion of the patients in the attacker's background knowledge that are correctly guessed from available probes. This means the fraction of patients correctly assigned by an attack $f$ to at least one of their probes and thus potentially compromised.

**Definition of the metric for deterministic attacks.** Consider a set $\mathcal{H}$ of *patients* and a set $\mathcal{P}$ of data *probes* that the attacker is trying to map to the patients in $\mathcal{H}$. Given a patient $h \in \mathcal{H}$, the attacker may acquire a probe belonging to the patient. We define the *probe ground truth G* as a function which assigns a set of possible probes $G(h)$ to every patient $h$.

Formally $G$ is defined as $G : \mathcal{H} \to 2^{\mathcal{P}}$, where $2^{\mathcal{P}}$ is the set of all subsets, so called powerset, of $\mathcal{P}$. In our case $G(h)$ consists of all publicly available WSIs of the patient $h$. The probe ground truth is not known to the attacker.

Consider an *attack $f : \mathcal{P} \to \mathcal{H}$* on the identity of the patient $h$ using the probe $p$. That is, given a probe $p$, the attack $f$ assigns a patient $f(p) \in \mathcal{H}$ to the probe $p$. We say that a given patient $h \in \mathcal{H}$ is *f-vulnerable* if there is a probe $p \in G(h)$ such that $h = f(p)$; that is, if at least one of the probes of the patient $h$ is correctly assigned to $h$ by the attack.

We define *worst-case probe attack success rate* by

$$R_s(f) = \frac{\text{number of } f-\text{vulnerable patients}}{|\mathcal{H}|} \qquad (1)$$

i.e., ratio of number of $f$-vulnerable patients to total number of patients. This equivalent to the common definition of success rate as defined in[23].

Observe that there is always a perfect attack $f$ under which all patients are $f$-vulnerable: a perfect attack simply assigns correctly a patient to at least one of his/her probes. However, such an attack cannot be practically implemented as attackers typically have limited background knowledge. So we consider families of possible attacks, an *attack domain $\mathcal{F}$*, based on the type of the background knowledge and algorithms used for the probe-patient assignment. In this work we consider attack domain $\mathcal{F}$, in which attacks map probes to patients according to the WSI similarity. However, the above defined metric can be understood in a more general sense as a method of measuring an attack on partially anonymized data sets using publicly available data.

**Application to WSI linkability risks.** In this paper we apply the metric to the analysis of linking risks related to WSI data. The attacks $f \in \mathcal{F}$ are implemented by various types of algorithms that are able to deterministically assign WSI probes to patients with WSIs in the attacker's background knowledge utilizing image similarity (attack domain $\mathcal{F}$). The analysis assumes the attacker does not have any explicit information about the mapping of the probes $\mathcal{P}$ to patients in $\mathcal{H}$. The resulting $R_s$ is a proportion of patients for which given attack $f$ is able to link WSI from the probes to the WSI from background knowledge.

Note that an attack $f : \mathcal{P} \to \mathcal{H}$ is determined by three components: set of patients in attacker's background knowledge $\mathcal{H}$, set of probes $\mathcal{P}$, and set of algorithms implementing the assignment of probes to patients. From a practical perspective, in order to analyze the risks we need to establish relationship between WSIs which might appear in the attacker's background knowledge and in set of probes.

Now note given the large size and detailed tissue structure of WSIs, it is reasonable to expect that if an identical WSI appears in both sets, a trivial bit-by-bit comparison algorithm would be able to assign this probe to the correct patient deterministically and unambiguously. Comparing larger data sets can be made more efficient by using cryptographic hash functions on each WSI and comparing resulting hashes. But to what extent does this apply also to other WSIs related by spatial or temporal relation to a single patient? In order to have basis for such analysis, we develop a taxonomy of WSI relations in WSI hierarchy.

**Possible model extensions.** The above metric can also be generalized in several ways. (a) One may consider randomized attacks, where the attacker no longer deterministically assigns probes to patients but may instead employ randomness in the choice: from random or fixed assignments of probes to patients without considering the content of each probe (probably not really useful except very small data sets with very privacy-threatening background knowledge) to any type of randomized algorithm. In such a case we would measure the worst-case probability (i.e., highest possible probability) of hitting the right

patient. (b) Patients may be assigned prior probabilities (weights) based on their priority/availability (e.g., more vulnerable patients might be prioritized by the attack model designer, such as persons of public interest, for whom the attack is causing more harm); that is, define $\pi_h \in [0, 1]$ for each patient $h \in \mathcal{H}$ so that $\sum_{h \in \mathcal{H}} \pi_h = 1$, and then define $R_s(f)$ to be the sum of probabilities of all $f$-vulnerable patients. In our basic definition above, the probabilities are uniform—that is, $\pi_h = 1/|\mathcal{H}|$ for every $h \in \mathcal{H}$. (c) Probes in $G(h)$ may be assigned probabilities of being revealed to the attacker. Our basic definition of $R_s(f)$ would then be the expected number of $f$-vulnerable patients when a probe is chosen randomly for each patient. Presence of patient prior $\pi_h$ allows us to express additional knowledge about the patients. For instance if the attacker can improve patient estimates based on their geographical location: e.g., due to source hospital being in attacker's background knowledge and the probes coming from a known hospital, we can introduce patient prior based on the match of source hospitals to model increased vulnerability of patients with matching hospital.

This extension maps to the formal frameworks of Quantitative Information Flow[24,25], where the secret is $\mathcal{X} = \{(h_1, p_a), (h_2, p_b), \dots\}$ with the prior distributions, the communication channel $C$ is the release of a probe $p \in \mathcal{P}$, i.e., $C(h, p) = p$, and the attacker is trying to guess the secret $(h, p)$ using a possibly randomized algorithm, which may utilize his background knowledge.

The analysis can be extended to other types of data. The background information as well as the probes can contain additional data types—e.g., phenotypic, clinical, omics, or other types of imaging. The attack model allows describing this as the attack domain $\mathcal{F}$ can contain attack algorithms utilizing these additional data types. However, these attacks have been studied elsewhere in the literature and are not subject of this paper, which focuses on developing recommendations for WSI data.

Furthermore, when considering WSIs from the privacy risk perspective, it is useful to consider them in terms of the conditions required for any variable to be considered an identifier[26]: distinguishability, replicability, and availability.

*Distinguishability* refers to the ability of the data to act as a fingerprint distinguishing the donor. One WSI is always distinguishing using direct bit-by-bit comparison; hence when the same WSI appears in two data sets, they are directly linkable. For different WSIs with increasing distance in the hierarchy (H-1) → (H-5) and to lesser extent also (H-a) → (H-c), each one of them is still individually distinguishing, but their linkability may decrease. The decrease in linkability between the (H-1) and (H-2) classes is demonstrated in the experimental part of this paper.

*Replicability* refers to what extent one can reproduce the data—either in succession or, more importantly, in more distant points in time. For WSIs the replicability is likely to decrease rapidly with increasing distance in spatio-temporal hierarchy. If the same slide is physically available (H-1), it can be scanned again, resulting in a WSI very similar to the previously obtained WSI from the same slide, with small differences related to aging of the slide (fading of staining), differences in physical preparation for scanning (e.g., dust particles), and properties of the scanner. Other means to replicability imply producing multiple very similar slides from different biological material. If different slides from the same block of tissue are used (H-2), the biological structures are not identical and the near perfect replicability is impossible. However, even imperfect replicability can still lead to linkability, as demonstrated later in this paper. For more distant temporal relation (H-5), replicability is unlikely due to the previous material being fixated in the slide preparation process, while new biological material remaining in the patient's body is subject to further biological development (e.g., natural growth, shrinkage/dying as reaction to treatment), making new samples diverge from previous ones. In addition, regarding tumor tissue, this is intentionally removed during surgery, hence new relevant material is no longer available in the patient's body subsequent slide replication.

*Availability* (also sometimes denoted as "knowability" in European Medical Agency guidelines[27]) describes the extent to which the data is accessible to potential attackers. Availability of WSI data has been evolving recently: a decade ago, the WSI was mostly limited to diagnostic purposes, if available as digital imaging at all. Some digital imaging has been used for education and training of medical professionals[28–30], but not to the extent of substantially increasing likelihood of identification except for diseases that are so rare that they already identify the patient. The above mentioned CAMELYON competitions demonstrated, however, how the situation is changing due to the demands and hopes put into development of novel AI methods to support cancer diagnosis and treatment; data are becoming rapidly available either under contracts or even as open data sets, thus substantially increasing availability. More recently, some publicly available data sets, such as PCam[31], have been made available as (labeled) tiles, instead of full WSI. This was done primarily to simplify training in tile-based AI models, but as a side effect it can also decrease linking risks if only limited number of tiles is released from each contributing WSI. Tile-based approaches are also considered in the experimental evaluation below.

## Design of experimental evaluation of WSIs linking risks

**The attacker model implementation.** We consider attackers assigning WSI probes to a patient's WSIs using similarity measures on features extracted from WSIs using common deep learning methods. Specifically, we assume that the attacker is given a WSI probe $p$ (i.e., probes consisting of WSI data only) and a background knowledge of WSIs $b_1, \dots, b_n \in \mathcal{B}$ associated to the patients $h_1, \dots, h_n$, respectively. Then the attacker proceeds as follows:

- Apply feature extractor to all WSIs, obtaining feature vectors $w[p], w[b_1], \dots, w[b_n] \in \mathbb{R}^k$ corresponding to the WSIs. Here $k$ is the number of features extracted from each WSI and is typically in the thousands.
- Apply similarity measure M to all pairs of vectors $w[p]$ and $w[b_i]$, obtaining $M(w[p], w[b_i])$ for all $1 \le i \le n$.
- Assign the WSI probe $p$ to the patient with the WSI $b_i$ with the maximum $M(w[p], w[b_i])$ among all $b_1, \dots, b_n$, i.e., such that

$$i \in \arg\max_j \left\{ M(w[p], w[b_j]) \,|\, j = 1, \dots, n \right\} \qquad (2)$$

In case that arg max contains more than one index, we select the smallest one.

The implementation of the attack model is based on the "prosecutor model"[18,32]—i.e., attacker knowing or assuming that the patient is in the background.

**Feature extraction.** The purpose of the feature extraction is to squeeze the high-dimensional WSIs into numerical vectors of smaller dimension. A conventional approach is to use a trained deep neural network model which takes images as inputs and outputs their vector representations, while preserving important features of the WSIs. To implement the feature extraction and to demonstrate variability in efficacy between different extractors we consider the following neural network models pre-trained on the ImageNet data set[33]:

- ResNet[34] – output dimension 2048 features;
- VGG16[35] – output dimension 25,088 features;
- Inception – output dimension 51,200 features;
- Img2vec[36] – a Python package that uses a pre-trained ResNet network to extract features of the dimension 512;
- SimCLRv2[37] – fine-tuned model on 100% of labels with output dimension of 8192 features;



**Fig. 10 | Example of probe selection for (E-2b) with minimum distance threshold *l* = 4 mm.** A pivot slide (in the centre) is selected as background knowledge. Additional slides from the same tissue block at a distance of 3 mm from the pivot are removed as they are not further than the threshold *l*, while other slides at 6 mm (i.e., 3 mm + 3 mm) are included in $\mathcal{P}$.

In addition, we have also included a specialized VGG16 feature extractor (denoted VGG16histo) that has been trained specifically for prostate cancer diagnosis[38] and achieves state-of-the-art diagnostic performance; this allows comparison of the generic ImageNet-trained VGG16 extractor with a very specialized extractor focused on detailed tissue structures to detect cancer patterns. These models from computer vision demonstrate their ability to relate slides which are nearby in the hierarchy, namely classes (H-1) and (H-2). In each of these cases, we remove the top layers of these networks, that originally solve image recognition problems, and use their internal representations of the input WSIs as feature vectors. Note that each WSI needs to be either downscaled or cropped to fit as an input of these networks, which is 224 px × 224 px. In our analysis we consider both alternatives and compare attacks using complete downscaled WSIs with attacks using WSIs cropped to their central parts. Also note that full resolution has not been used as the detailed structures are only similar on consecutive slides, i.e., (H-1) and consecutive slides of (H-2) using our hierarchy, for which the presented methods on down-scaled images are already very effective, as shown in Figs. 4 and 8.

**Similarity measures.** To measure similarity of feature vectors we use Cosine similarity

$$M_{\cos}(v,w) = \frac{\sum_{i=1}^{n} v_i w_i}{\sqrt{\sum_{i=1}^{n} v_i^2}\sqrt{\sum_{i=1}^{n} w_i^2}} \tag{3}$$

which is normalized in the range [−1; 1]. We have also implemented Euclidean metric ($M_{\mathrm{euc}}(v,w) = 1/(1 + \sum_{i=1}^{n}(v_i - w_i)^2)$) but the trends are generally the same and the Cosine similarity achieved slightly higher $R_s$ scores. Thus, in the evaluation we use Cosine similarity exclusively, as it provides the upper bound on vulnerability.

**Human samples**
Slides for the (H-1) hierarchy analysis were provided by Masaryk Memorial Cancer Institute (MMCI) and contained colorectal cancer WSIs, breast cancer WSIs, and prostate biopsies. Slides for all the other hierarchy analyses were prostate biopsies provided by Medical University Graz (MUG). All images were stained using hematoxyline & eosin staining in a single batch; therefore, the experimental analyses are restricted to the (H-a) step from the WSI staining hierarchy.

The research has been authorized under ethics vote at Masaryk Memorial Cancer Institute, no. 2021/3375/MOU (MOU 385 920), and at Medical University Graz, no. 1072/2019. Participats provided informed consent for the use of their samples. This work has not attempted to identify nor identified any data subjects, it solely performed the linking-based risk analysis on WSI data.

**Data sets**
The attack analyses focused primarily on linking complete WSIs and we performed experiments with the following data sets based on the WSI spatiotemporal hierarchy.

(1)  [Based on (H-1)] Consists of 28 slides from different patients with prostate, breast, or colorectal cancer provided by MMCI, which were scanned on Pannoramic® MIDI and Pannoramic® 250 Flash III scanners by 3DHistech (Budapest, Hungary) at a resolution of 0.172 μm/px (WSI sizes: 18.3 Gpx for Flash scanner and 23.3 Gpx for MIDI scanner). Of each pair, one WSI is randomly assigned to the probe set $\mathcal{P}$ and the other is assigned to $\mathcal{B}$ of background knowledge.

(2)  [Based on (H-2)] Consists of WSIs from consecutive and non-consecutive prostate biopsy slides provided by MUG, which were scanned on Aperio AT2 scanner at a resolution of 0.25 μm/px (WSI size of 9 Gpx). Note that in the following text we use the term "slides" for better readability, but precisely speaking the sets $\mathcal{B}$ and $\mathcal{P}$ are populated by the WSIs from the discussed slides.

 (1)  The whole data set consists of 151 pairs of (directly) consecutive WSIs from 151 different patients (see Fig. 1d). Pairs are visually similar, but there are also visible differences. One randomly selected slide from each pair goes into $\mathcal{B}$−the attacker's background knowledge−and the other into the probe set $\mathcal{P}$.

 (2)  Consists of 558 slides from 80 patients (average 6.975 slides per patient). Metadata on these slides contained information on order of cuts and approximate distance from the previous one (ranging 3 mm to 5 mm). We used this information to compose various data subsets to study the influence of the distance between slides in background knowledge and probes (see Fig. 10). For each threshold distance *l* and each patient, the sets $\mathcal{B}$ and $\mathcal{P}$ were populated as follows: a pivot slide is selected randomly from the patient and is inserted into $\mathcal{B}$; then, all the patient's slides that are at a distance greater than *l* from the pivot are added to the $\mathcal{P}$. Note that the set of slides added to $\mathcal{P}$ may be empty for some patients, if there are no slides available in distance greater than *l*; this practically happens for larger threshold distances. In such a case, the pivot slide in $\mathcal{B}$ for the given patient has no corresponding slides in $\mathcal{P}$.

We also study the ability to attack the data set when the overall shape of the tissue cannot contribute to the extracted features and hence to similarity. We cropped the WSIs so that the resulting image contained internal parts of the tissue only (further denoted as cropped WSIs). Registration (alignment) of the WSIs was done before the cropping, to model worst case scenario that the attacker can get access to spatially corresponding cropped WSIs (see Fig. 7a). We studied the influence of decreasing overlap of these cropping regions on $R_s$ too, by

shifting the crop regions by fixed amounts in random direction for each WSI pair (see Fig. 7b). These experiments were done for both consecutive slides (cE-2a) and non-consecutive slides (cE-2b).

## Statistics and reproducibility

We have tested attacks using all the deep learning models listed in Design of Experimental Evaluation of WSIs Linking Risks to extract feature vectors from the images. Moreover, two similarity metrics were tested to compare the extracted feature vectors: cosine and Euclidean. However, the trends produced by the two metrics are generally the same and the cosine similarity achieved slightly higher $R_s$ scores; therefore, we only present results using cosine similarity. The experiments (E-1) and (E-2a) were repeated 40 times. On the other hand, experiment (E-2b) was repeated 20 times, as this experiment has much higher computational requirements and 20 repetitions were sufficient to characterize the sample distribution of the observations for this case. All data sets are regenerated randomly from the available data for each run. Naturally, at each run we measure a different $R_s$ value, and the results presented in the graphs below show the characteristics of the obtained sample distributions of $R_s$ (avoiding distribution normality assumptions): median, quartiles, and min-max range using box plots, overlaid with bootstrap-based 95% confidence intervals (dashed tabs) and with jittered plots of actual data points, unless explicitly stated otherwise. Note that the box plots are modified so that the whiskers visualize min–max range without eliminating outliers.

## Reporting summary

Further information on research design is available in the Nature Portfolio Reporting Summary linked to this article.

## Data availability

Source data of Results is included with the paper (source-data.zip), which contains data in CSV format for each experiment, an R notebook to generate all the graphs presented in the paper, and an HTML version of the R notebook. The source WSI data used in this paper are available under restricted access for data protection resons, access can be requested via BBMRI-ERIC, a European Research Infrastructure, from MMCI via https://directory.bbmri-eric.eu/menu/main/app-molgenis-app-biobank-explorer#/collection/bbmri-eric:ID:CZ_MMCI:collection:LTS and MUG via https://directory.bbmri-eric.eu/menu/main/app-molgenis-app-biobank-explorer#/collection/bbmri-eric:ID:AT_MUG:collection:FFPEblocksCollection. The access request can be filed from these URLs via BBMRI-ERIC Negotiator. Expected time frame for data release, if approved, is 1–2 months. Source data are provided with this paper.

## Code availability

Code is available at GitHub at https://github.com/RationAI/WSI-anonymity.

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

## Acknowledgements

This work has been supported by EOSC-Life project supported by EU Horizon 2020, grant agreement no. 824087, as a part of WP1 Demonstrators under APPID 1228 "Cloudification of BBMRI-ERIC CRC-Cohort and its Digital Pathology Imaging"; by the Czech Ministry of Health (MMCI 00209805) and the Czech Ministry of Education, Youth and Sports (LM2018125 – BBMRI-CZ); the Austrian Federal Ministry for Education, Science and Research (BMBWF-10.470/0010-V3c/2018; BBMRI.at); the SVDC project (funded by the Sardinian Regional Authority); the Czech Science Foundation, Grant No. 21-24711S. Computational resources were supplied by the project 'e-Infrastruktura CZ' (e-INFRA LM2018140) provided within the Projects of Large Research, Development and Innovations Infrastructures program.

## Author contributions

P.H. has led the work, developed the concept of the paper, led development of the methodology, co-authored the attack model, evaluated experiments, and led development of guidelines, and led paper editing. T.Br. co-designed the methodology and is the main author of the formal attack model. H.M. and L.P. co-designed the methodology. T.Bi. has implemented experiments. R.N. and K.Z. contributed data for the experiments and contributed to the guidelines. M.P. contributed data for experiments. F.P. contributed to linking the model to the common privacy models and I.S. contributed alignment with GDPR.

## Competing interests

K.Z. is founder and CEO of Zatloukal Innovations GmbH. The remaining authors declare no other competing interests.
