## [Peer Review File · Nature Communications]

Privacy Risks of Whole-Slide Image Sharing in Digital PathologyREVIEWER COMMENTS

Reviewer #1 (Remarks to the Author):

The authors present the paper entitled "Privacy Risks of Whole-Slide Image Sharing in Digital Pathology" in which they presented a model for risk assessment of whole-slide images for particular well-defined scenarios under identity disclosure attacks. The authors assumed that the WSIs are available and they did not consider previous security layers of data encryption or secure channels of communications. The criterion adopted by the authors is also associated that it is possible to associate a probe WSI with other WSI from the background knowledge of the patients only using visual similarity through a deep-learning approach of feature extraction of those WSIs. Interesting findings are presented in the experimental design and results and some recommendations are presented considering different scenarios with WSIs of consecutive slides, spatial distance, cropped image regions, and so on. However, the authors suggest in subsection 2.2 a WSI hierarchy but the experiments and results did not consider all of these. Considering the pathologies associated with rare cases or low incidence, there is not too much discussion about it, which is a relevant topic. It lacks a statistical analysis to evaluate the results of each experiment considering the variability of the data and how a batch effect or data-dependency could be associated with the risks, even the quality of WSIs during the digitalization process and slide scanner used. Finally, the deep learning models used as feature extractors are coming from general-purpose and trained models with natural images for computer vision tasks. It is needed experiments using specific experiments using publicly available deep learning methods trained or fine-tuned with histopathology images. Mainly, there are some sentences without support in conclusions and suggestions to complement them. Hence, this is a very interesting and original work about a not too much-discussed topic of the privacy risks of WSIs in digital pathology in the current deluge of open and publicly available data and rising digital pathology information systems and applications, but it requires more experimental results with an appropriate validations scheme based on statistical analysis to strengthen the conclusions, despite that the suggested guidelines are very useful and valuable for a more open discussion and regulation. In order to have a complete privacy risks analysis of WSIs in digital pathology, it is important not to unattached the technology platforms and infrastructure of this discussion.

Reviewer #2 (Remarks to the Author):

The paper addresses an important problem that has not been adequately studied in the literature.

However, in my opinion, some shortcomings stand in the way of publication.

The authors analyse only "half of the problem" in all experimental work by evaluating metrics for "positive pairs."

Imagine you have a CNN that outputs a constant vector representation, regardless of the WSI you input as input (not a very useful CNN).

The metrics for this CNN in Tables 1 through 5 would be 1 for cosine and infinity for Euclidean similarity. One might conclude that there is a high risk of linking WSI, etc. However, this would not be the case. The number of false negatives would also be high.

I missed the discussion of the 'other part' of false negatives. Without this part, it is impossible to determine whether the risk is high or not, whether the similarities are high or not, etc.

Additional comments:

-The authors could better justify the need to introduce new metrics and definitions by mentioning some of the literature and explaining why they are not applicable.

-On page 10, 3rd item in the list, the formula seems just plain wrong. I think it would make more sense (in latex):

$$b_i = \arg\max_j \{ M(w[p], w[b_j]) \mid j = 1, \dots, n \}$$

-WSI were scaled to 224x224. This is very small. Would not all slides look the same, artificially increasing the similarity metric?
- Euclidean similarity. If the distance is zero, the similarity is infinite. It is common to assume $M = \frac{1}{1+\text{distance}^2}$, where the similarity is now limited to 1.

-When evaluating, is the method able to tell that the probe is not in the background? That would make it more realistic.

- Results: Why do not you also give the standard deviation of the 20 measurements?

- End of Section 3.1: Please comment on the results for Figure 8b to help the reader read the figure.

Reviewer #3 (Remarks to the Author):

Summary:

The paper performs a privacy risk analysis of sharing whole-slide image (WSI) data widely used in digital pathology for diagnosis and machine learning model training. In particular, the authors develop a model that estimates the privacy risk via identity disclosure attacks, i.e., by computing how many patients in a WSI dataset can be successfully identified (after this has been shared) by an adversary with access to some partial knowledge about it. The proposed model is implemented following a methodology based on feature extraction via deep learning and similarity matching. Using real-world WSI data the authors experimentally evaluate how factors such as spatio-temporal proximity and staining methods affect the privacy risk and then use their results to develop guidelines for WSI data sharing with low privacy risks.

Comments:

The paper is interesting and timely as WSI data sharing is crucial in digital pathology for developing better insights about diseases and building (machine learning) tools that can assist medical doctors in their diagnosis. To the best of this reviewer's knowledge, this is the first paper that analyzes the privacy risks associated with WSI data sharing and can be a milestone that will likely spur further research on the topic. Moreover, the paper's methodology is reasonable and serves well the purpose of demonstrating the privacy risks of WSIs in an easy-to-understand manner that even non-expert audience would grasp. Other strong points of the paper include the usage of real-world WSI data for the experimental evaluation, the extensive analysis of factors (e.g., spatio-temporal proximity and staining methods) that affect the patient identification, as well as the proposal of guidelines for minimizing privacy risks when sharing WSI data. Nonetheless, this reviewer has a few concerns which are discussed below.

In particular, this reviewer is concerned about the estimation of the privacy risk by the proposed model: The computed risk is conditional to the presence of a patient data both in the attacker's background knowledge and the patient dataset to be shared. First, this assumption might be deemed as unrealistic depending on the setting and the authors should describe in more clarity the envisioned attackers, i.e., who they might be, how they might have access to this side knowledge (and how this is linked to the WSI hierarchy described), and what would be their motivation for launching the attack. While this reviewer understands that the authors consider such an attacker to estimate sufficiently the privacy risk, a crucial second question is whether this attacker is the "worst-case" attacker or not and accordingly if their method yields some bound on privacy loss or possibly under-estimates it. For instance, the model defined for the attack is not based on some known privacy definition, e.g., differential privacy, k-anonymity or similar. Moreover, the authors leave the additional information that might be inferred from a WSI fingerprint as an open question: This reinforces the above concerns about under-estimating the privacy risk and not relying on formal definitions (e.g., differential privacy) that can possibly bound such arbitrary leakage. Finally, the privacy risk considered is deterministic as

it captures the number of patients uniquely re-identified (i.e., with probability 1) in the shared dataset. The authors should also take into account probabilistic adversaries (e.g., see Bayes vulnerability [1, 2]) which are likely to yield additional interesting results in terms of privacy risk.

While the methodology employed to instantiate the privacy model is simple and clear the paper does not reason about nor analyze sufficiently its effectiveness. For instance, it is unclear why the model is instantiated with two similarity metrics (cosine vs. euclidean) for the matching purpose: the paper's results follow a similar trend with both metrics so a reader is left wondering why both metrics are important or which one to choose for future evaluations. Similarly, there is no explanation why five different feature extractors are used: Is one approach better than the rest and why is that the case? While there does not seem to be a connection between the privacy risk estimation and the number of features extracted (\$k\$) the authors should analyze what qualitative properties of the deep learning models used are important for realizing the proposed framework and why would they be suitable for WSIs in particular. Overall, the methodology could benefit from such a rigorous analysis to convince the reader about the experimental choices made.

The last part of the paper proposing guidelines for low-risk sharing of WSI data is interesting and important. However, a reader would expect to find an explicit connection between the proposed guidelines and the experimental results of the paper along with the threat model assumptions that justify the result. Moreover, the authors define an epsilon-safe dataset as part of their model definition, but this is nowhere used nor discussed in these guidelines. Could this privacy definition be used in practice as a technical tool to release a dataset with low privacy risk? If yes, how would someone tune the epsilon parameter and if not, why so? Additionally, it is unclear if the proposed privacy model is accompanied by an open-source tool that practitioners could make use of to aid in their decisions of sharing WSI data? Finally, the guidelines proposed by the authors seem to be missing an important component: Where is the patient consent involved in the data sharing process? Should not the patients be informed about the privacy risks estimated and make a decision about their WSI data being shared?

More detailed comments:

- Section 2.3: In the proposed methodology it seems that the probe needs to be of the same size as the WSI in the shared dataset. This reviewer is wondering if, e.g., substantially smaller probes (in terms of # of pixels) could be sufficient to identify a patient as long as they are derived from a point-of-interest with discriminating information about the corresponding patients. It would be interesting to see an experimental analysis on this issue.
- Section 2.3.2, Setting E-2a: are these 158 pairs taken from one patient or different ones?
- Section 2.3.2: It is unfortunate that settings H-4 and H-5 are not evaluated. In this reviewer's opinion they could yield interesting results.
- Table 1: Please explain what the 'autoencoders' column indicates. Is it another feature extraction approach tested but not described in the main part of the paper?
- Figure 8: Why is the y-axis label named 'score'? Shouldn't it be V_{att} ?
- Table 4: Please indicate the actual overlap (in percentage) of the WSIs to get a better understanding of the attack and its results.

Overall, while the paper is well-written and easy to follow there exist some typos that should be fixed. Some examples:

Typos:

Section 1: "[...] known as digital pathology.[2]"

Section 1: "[...] in the diagnostic process.[3]"

Section 1: "[...] digitized using an slide scanner [...]"

Section 1: "[...] but does necessarily appear in the dataset [...]"

Section 2.3.1: "[...] WSI probes to a patients' WSIs [...]"

Section 4: "[...] caution needs to exercised to mitigate [...]"

References:

[1] G. Smith, On the foundations of quantitative information flow. In: International Conference on Foundations of Software Science and Computational Structures. Springer, Berlin, Heidelberg, 2009. S. 288-302.

[2] M. Alvim, et al. The Science of Quantitative Information Flow. Springer International Publishing, 2020.

Response to Reviewer #1

Reviewer #1

The authors present the paper entitled “Privacy Risks of Whole-Slide Image Sharing in Digital Pathology” in which they presented a model for risk assessment of whole-slide images for particular well-defined scenarios under identity disclosure attacks. The authors assumed that the WSIs are available and they did not consider previous security layers of data encryption or secure channels of communications. The criterion adopted by the authors is also associated that it is possible to associate a probe WSI with other WSI from the background knowledge of the patients only using visual similarity through a deep-learning approach of feature extraction of those WSIs. Interesting findings are presented in the experimental design and results and some recommendations are presented considering different scenarios with WSIs of consecutive slides, spatial distance, cropped image regions, and so on.

Thank you for the encouragement.

In the introduction we mention that practices for sharing WSIs vary considerably, from sharing it as personal data under a contract to considering WSIs fully anonymous a sharing it without any conditions or restrictions. When the data is shared under a contract (i.e., organizational measures are in place) and secure storage and communication channels are used (so, technical measures are in place), the risks are typically managed by means analogous to any other personal/sensitive data types used for [medical] research purposes.

The combination of organizational and technical measures is described in the introduction of Section 4.1 and we have added another mention of it in the guidelines for pseudonymized data in Section 4.1. Please note that the same section also recommends application of lightweight measures for processing de facto anonymous data to mitigate residual risks.

Reviewer #1

However, the authors suggest in subsection 2.2 a WSI hierarchy but the experiments and results did not consider all of these.

This paper is clearly the start of the process. We have limited ourselves to the similarity classes where the likelihood of linking or re-identification attacks is highest, and we have observed a significant drop in the ability to link WSIs with increasing spatial distance. The methods we tested in our experiments started having poor success rates already at the level of sufficiently distant slides in the same tissue blocks (i.e., the (H-2) level). Hence, at this stage, we have not proceeded further with down the hierarchy for the experimental analysis. An explanation has been added to the paper.

We expect this paper to have follow-up publications where others can analyze risks across more distant classes if stronger attack algorithms are developed and we ourselves aim to organize an international competition to explore this, as we state in the Conclusions section.

Reviewer #1

Considering the pathologies associated with rare cases or low incidence, there is not too much discussion about it, which is a relevant topic.

We have added additional recommendation for small (sub)populations (recommendation 6), plus a paragraph (marked as “Note”) has been added to the recommendation 5 on data derivation. Note that this is a general topic, which has to be considered by any risk assessment methodology and not very specific to whole-slide images (WSIs).

Reviewer #1

It lacks a statistical analysis to evaluate the results of each experiment considering the variability of the data and how a batch effect or data-dependency could be associated with the risks, even the quality of WSIs during the digitalization process and slide scanner used.

We are not certain how this comment was meant. However, we have reworked presentation of the results into a form which we hope gives much better insights into the variability of the obtained data. The averages/medians have been replaced by box plots overlaid by jittered raw data, so that the readers can assess both the overall distributions.

Dependencies of the WSI data has been explored using the (H-*) hierarchy defined in the paper. We admit that not all the steps of the hierarchy were explored and some are left for future work once stronger methods are available, as the methods used in the paper showed big drop in R_s already on the more distant WSIs in the (H-2) step.

Reviewer #1

Finally, the deep learning models used as feature extractors are coming from general-purpose and trained models with natural images for computer vision tasks. It is needed experiments using specific experiments using publicly available deep learning methods trained or fine-tuned with histopathology images.

We have of course tested feature extractors based on histopathology-optimized models,¹ but these generally delivered worse results and haven't been included in the final selection of feature extractors into the paper.

¹ <https://doi.org/10.1101/2022.03.31.486599>

Upon closer inspection, the main reason seemed to be that these methods focus on fine details in the tissue structures, while for (re)identification or linking of WSIs are much more important the overall shape and larger structural elements, and thus we were already successful with applying general purpose computer vision models.

The re-identification might be of course further attempted to be improved by training very specialized models that are able to use both the larger structural elements and the fine-grained tissue structures. This is one of the research directions we aim to pursue in the future.

Reviewer #1

Mainly, there are some sentences without support in conclusions and suggestions to complement them.

We have reviewed the Conclusions section. Namely each of the recommendations has been extended with a Rationale, linking it to the experimental results as appropriate.

Reviewer #1

Hence, this is a very interesting and original work about a not too much-discussed topic of the privacy risks of WSIs in digital pathology in the current deluge of open and publicly available data and rising digital pathology information systems and applications, but it requires more experimental results with an appropriate validations scheme based on statistical analysis to strengthen the conclusions, despite that the suggested guidelines are very useful and valuable for a more open discussion and regulation. In order to have a complete privacy risks analysis of WSIs in digital pathology, it is important not to unattached the technology platforms and infrastructure of this discussion.

Response to Reviewer #2

Reviewer #2

The paper addresses an important problem that has not been adequately studied in the literature.

Reviewer #2

However, in my opinion, some shortcomings stand in the way of publication. The authors analyse only “half of the problem” in all experimental work by evaluating metrics for “positive pairs.”

Imagine you have a CNN that outputs a constant vector representation, regardless of the WSI you input as input (not a very useful CNN). The metrics for this CNN in Tables 1 through 5 would be 1 for cosine and infinity for Euclidean similarity. One might conclude that there is a high risk of linking WSI, etc. However, this would not be the case. The number of false negatives would also be high. I missed the discussion of the 'other part' of false negatives. Without this part, it is impossible to determine whether the risk is high or not, whether the similarities are high or not, etc.

-On page 10, 3rd item in the list, the formula seems just plain wrong. I think it would make more sense (in latex):

$$b_i = \arg \max_j \{M(w[p], w[b_j]) | j = 1, \dots, n\}$$

We have realized there is one element missing in the description of our experimental work – which we realized when you gave us the example with the constant output of the CNN. The model has been made on the assumption of ‘normal’ CNNs that are very unlikely to give identical result for different slides. The missing part is: *should such situation happen and arg max returns more than one index, we select the lowest index* – this has been our default selection rule, as the distance metrics do not give us any further information. Note that this has never happened in practice and we always had a single maximum.

We have also fixed the formula – thank you for noticing the problem!

A side note: Originally, the paper had been started with defining the metric using randomized algorithms, which would be more appropriate to link the model to the game theory models. However, the feedback we got from our colleagues who are applied privacy researchers, is that intricacies of such a model are very hard to understand. Thus the model has been reformulated as a deterministic. The randomized model would be more elegant here as the selection of matching patients would be done randomly.

The ‘false negatives’ part is actually already captured in the $R_s(f)$ model:

$$R_s(f) = \frac{\text{number of } f\text{-vulnerable patients}}{|\mathcal{H}|}$$

The $|\mathcal{H}|$ contains all patients and hence the false negatives are also captured in the denominator of the metric. In the case of the constant output of the CNN, the model would give the ratio between the correctly matched WSIs and all the patients.

Reviewer #2

Additional comments: -The authors could better justify the need to introduce new metrics and definitions by mentioning some of the literature and explaining why they are not applicable.

We believe that this is a standard evaluation of algorithm accuracy. This is not a novelty of the paper, but we define it only to have a rigorous formulation of the success rate for further work in the paper. The success rate is equivalent to the success rate defined in [1] and success rate for prosecutor model in [2, 3]. We have changed the name and notation (from V_{att} to R_s) to make it clearer and made the references to these works. See also our response to Reviewer #3.

Reviewer #2

-WSI were scaled to 224x224. This is very small. Would not all slides look the same, artificially increasing the similarity metric?

If all slides were looking the same, it would be actually giving disadvantage to the models we have applied – because in order to assign slides between \mathcal{B} and \mathcal{D} , one needs to be able to distinguish among the slides. Increasing resolution of the image would mainly make sense if the models were unable to assign slides correctly at least for the Tables 1 and 2. Since we have practically demonstrated that the slides are already sufficiently distinct at this resolution, we consider the 224 px \times 224 px sufficient for the experiments. A note has been added to Section 2.3.1 to clarify this.

Using full resolution of the WSIs, i.e., 9 Gpx to 23 Gpx in our case, requires accessing the WSIs on per-tile basis and then integrating the feature vectors in a way that is robust with respect to the registration imperfections (in case of same or consecutive slides) and generally dealing with registration or translational/rotational invariance of this process in case of more distant slides, where the registration is much more difficult and less reliable. This could be relevant for further experiments that we propose to analyze in the international competition.

Reviewer #2

- Euclidean similarity. If the distance is zero, the similarity is infinite. It is common to assume $M = \frac{1}{1+distance^2}$, where the similarity is now limited to 1.

Thank you for the suggestion, we have implemented it. However, for the reason discussed below in response to Reviewer #3 on using different metrics in the paper, we have resorted to using cosine distance only and we mention now in the paper that this modified Euclidean distance has been also evaluated with similar results.

Reviewer #2

-When evaluating, is the method able to tell that the probe is not in the background? That would make it more realistic.

There are two possible interpretations of this question:

(1) Whether the method is able to tell that the patient with the probe is not in the background.

We agree with the reviewer that this might be more realistic, but we have decided not to include this for the moment. We have modeled the worst-case scenario, where the attacker knows that a patient, to which the examined probe belongs, is in the background set \mathcal{B} —this is so called prosecutor model [2, 3], reference to which has been added to the paper.

When this condition is relaxed, the attack model is still possible but it would need to set a threshold under which the patient is considered not being present in \mathcal{B} . Exploration of setting such a threshold has been left for possible future work.

(2) If the question was meant as whether the method is able to tell that the given probe is itself in the background or not, this can be trivially done by bit-by-bit matching of the probe to all the WSIs in the background knowledge.

Reviewer #2

- Results: Why do not you also give the standard deviation of the 20 measurements?

We have reimplemented presentation of the results: for all the experiments except for (E-1), we are displaying box plots overlaid with the actual data using jitterplots, so that the characteristics of the distributions are readable from the graphs (after sufficient zoom in some cases) and the reader can also see the discrete steps of the R_s value distribution. For the (E-1) due to relatively low number of cases, we show histograms with the number of mismatched slides, hence providing overview of the complete data.

Reviewer #2

- End of Section 3.1: Please comment on the results for Figure 8b to help the reader read the figure.

Caption of the Figure 8 has been expanded, we hope it clarifies the situation.

Response to Reviewer #3

Reviewer #3

Summary:

The paper performs a privacy risk analysis of sharing whole-slide image (WSI) data widely used in digital pathology for diagnosis and machine learning model training. In particular, the authors develop a model that estimates the privacy risk via identity disclosure attacks, i.e., by computing how many patients in a WSI dataset can be successfully identified (after this has been shared) by an adversary with access to some partial knowledge about it. The proposed model is implemented following a methodology based on feature extraction via deep learning and similarity matching. Using real-world WSI

data the authors experimentally evaluate how factors such as spatio-temporal proximity and staining methods affect the privacy risk and then use their results to develop guidelines for WSI data sharing with low privacy risks.

Comments:

The paper is interesting and timely as WSI data sharing is crucial in digital pathology for developing better insights about diseases and building (machine learning) tools that can assist medical doctors in their diagnosis. To the best of this reviewer's knowledge, this is the first paper that analyzes the privacy risks associated with WSI data sharing and can be a milestone that will likely spur further research on the topic. Moreover, the paper's methodology is reasonable and serves well the purpose of demonstrating the privacy risks of WSIs in an easy-to-understand manner that even non-expert audience would grasp. Other strong points of the paper include the usage of real-world WSI data for the experimental evaluation, the extensive analysis of factors (e.g., spatio-temporal proximity and staining methods) that affect the patient identification, as well as the proposal of guidelines for minimizing privacy risks when sharing WSI data. Nonetheless, this reviewer has a few concerns which are discussed below.

Thank you for the encouragement!

Reviewer #3

In particular, this reviewer is concerned about the estimation of the privacy risk by the proposed model: The computed risk is conditional to the presence of a patient data both in the attacker's background knowledge and the patient dataset to be shared. First, this assumption might be deemed as unrealistic depending on the setting and the authors should describe in more clarity the envisioned attackers, i.e., who they might be, how they might have access to this side knowledge (and how this is linked to the WSI hierarchy described), and what would be their motivation for launching the attack. While this reviewer understands that the authors consider such an attacker to estimate sufficiently the privacy risk, a crucial second question is whether this attacker is the "worst-case" attacker or not and accordingly if their method yields some bound on privacy loss or possibly under-estimates it.

We have modeled the worst-case scenario, where the attacker knows that a patient, to which the examined probe belongs, is in the background set \mathcal{B} —this is so called prosecutor model [2, 3], reference which has been added to the paper. We consider this as a forward looking approach since as more and more WSI data sets become available, each linked with different background knowledge, it might be easier for the attacker to guess or assume that the patient is in the given data set based on the locality and other inclusion criteria of the cohort/dataset. This aspect should be considered in current projects and becomes particularly important when dealing with small populations – again, this has been added to the guidelines in the concluding section of the paper.

When this condition is relaxed, the attack model is still possible but it would need to set a threshold under which the patient is considered not being present in \mathcal{B} . Exploration of setting such a threshold has been left for possible future work.

We have also added a new paragraph into the “Problem statement” section exemplifying a simple attack and hopefully providing better context for the assumptions we are making in the model.

Reviewer #3

For instance, the model defined for the attack is not based on some known privacy definition, e.g., differential privacy, k-anonymity or similar. Moreover, the authors leave the additional information that might be inferred from a WSI fingerprint as an open question: This reinforces the above concerns about under-estimating the privacy risk and not relying on formal definitions (e.g., differential privacy) that can possibly bound such arbitrary leakage.

Our metric measures the success rate of a given attack, which is a natural metric often used in various safety settings, such as to measure quality of adversarial attacks on machine learning systems, or in cryptography to evaluate various attacks on a cryptographic protocols. It is equivalent to the definition of success rate in the highly-cited [1] (“The success rate τ is defined as the percentage of adversarial samples that were successfully classified by the DNN as the adversarial target class.”) and analogous to less rigorous definitions in other works [4, 5]. We have added a reference to [1] into the paper and renamed our metric from V_{att} (which was based on the background of the model in security games) to R_s .

Of course, the assurance of overall safety depends on the generality of the considered attacks, which we leave as a hyper-parameter in the form of the class of attacks. We think this agrees with the usual approach – e.g., in cryptography, where typically only computationally feasible attacks are considered. Moreover, showing that a dataset is not safe boils down to showing a particular type of attack that may violate privacy, which is the main concern of our experiments.

As for the other information that might be inferred from WSI: we are aware of this issue and it is covered in the final section of the paper and considered in the developed guidelines. But as these are covered by other studies (e.g., on genomics data privacy, if WSIs were used to infer mutations), we have decided to concentrate on a single well-defined problem of patient identification using the WSI as such, which is the main missing aspect in the literature. Further development towards measuring the amount of information that might be inferred from given data and combining it with the patient identification will be considered in future work.

Reviewer #3

Finally, the privacy risk considered is deterministic as it captures the number of patients uniquely re-identified (i.e., with probability 1) in the shared dataset. The authors should

also take into account probabilistic adversaries (e.g., see Bayes vulnerability [1, 2]) which are likely to yield additional interesting results in terms of privacy risk.

We had already considered this aspect and we even had text written for it, but in the end decided to remove it before submissions for the sake of simplicity. The text has been reverted and is now available as “Possible model extensions” at the end of Section 2.1.

We retain, however, the basic deterministic formulation of the attack model as the core of the paper as it is how the experiments have been implemented. The probabilistic extensions have also been difficult to understand for non-specialists when we tested their understandability with some of our colleagues.

Reviewer #3

While the methodology employed to instantiate the privacy model is simple and clear the paper does not reason about nor analyze sufficiently its effectiveness. For instance, it is unclear why the model is instantiated with two similarity metrics (cosine vs. euclidean) for the matching purpose: the paper’s results follow a similar trend with both metrics so a reader is left wondering why both metrics are important or which one to choose for future evaluations.

We have also reimplemented the Euclidean metric as per suggestion of Reviewer #2, obtaining similar results again, so we concluded that cosine distance is sufficient as it gives consistently highest R_s values and hence upper bound on vulnerability. We have simplified the presentation of results using cosine distance only. We have also modified the method description to mention that we have also implemented Euclidean similarity but not present the detailed results in the paper, as they are analogous.

Reviewer #3

Similarly, there is no explanation why five different feature extractors are used: Is one approach better than the rest and why is that the case? While there does not seem to be a connection between the privacy risk estimation and the number of features extracted (k) the authors should analyze what qualitative properties of the deep learning models used are important for realizing the proposed framework and why would they be suitable for WSIs in particular. Overall, the methodology could benefit from such a rigorous analysis to convince the reader about the experimental choices made.

We have employed state-of-the-art models trained on ImageNet for generic computer vision and a selection of models has been presented to demonstrate the range of their performance (which shows that, except for Inception, all other models can be successfully used for the attack). We have also experimented with other models – namely those that perform very well for histopathological diagnoses (e.g., VGG16-based prostate and breast cancer models),

however these provided performance systematically inferior performance to the models presented in the paper. This is due to the fact that the diagnostic models are using very fine textures in the native resolution images using patches of images and not full WSIs on its input – and these fine textures were not present in the downscaled WSIs. We have added a note on the selection of the feature extractors to the paper.

However, we expect that models can be actually optimized for the linking attacks and this is the main reason why we propose to organize the international competition. Once optimized models are available, we agree it would make sense to attempt to explain their behavior and analyze to what image features they are sensitive.

Reviewer #3

The last part of the paper proposing guidelines for low-risk sharing of WSI data is interesting and important. However, a reader would expect to find an explicit connection between the proposed guidelines and the experimental results of the paper along with the threat model assumptions that justify the result.

We have reviewed the Conclusions section. Namely each of the recommendations has been extended with a Rationale, linking it to the experimental results as appropriate.

Reviewer #3

Moreover, the authors define an epsilon-safe dataset as part of their model definition, but this is nowhere used nor discussed in these guidelines. Could this privacy definition be used in practice as a technical tool to release a dataset with low privacy risk? If yes, how would someone tune the epsilon parameter and if not, why so?

The ϵ -safety has been removed from the paper, as it was meant for the future use of theoretical reasoning about the attack domains and is not directly useful in the discussion of results as they are presented in the paper.

Reviewer #3

Additionally, it is unclear if the proposed privacy model is accompanied by an open-source tool that practitioners could make use of to aid in their decisions of sharing WSI data?

We are providing open-source implementation of the methods used in our experiments. This is a research toolbox and not a tool that is optimized to be easily adopted by non-experts. Nevertheless it is possible to adapt it for use with other people's data for their own analysis, too.

Reviewer #3

Finally, the guidelines proposed by the authors seem to be missing an important component: Where is the patient consent involved in the data sharing process? Should not the patients be informed about the privacy risks estimated and make a decision about their WSI data being shared?

Yes, and we considered this as part of the standard procedure that needs to be done for any release of personal data or anonymization of data (which is processing of personal data as well, and thus needs to be authorized and must have a legal basis under GDPR). A new paragraph has been added to the Section 4.1 to clarify this.

Reviewer #3

More detailed comments:

- Section 2.3: In the proposed methodology it seems that the probe needs to be of the same size as the WSI in the shared dataset. This reviewer is wondering if, e.g., substantially smaller probes (in terms of # of pixels) could be sufficient to identify a patient as long as they are derived from a point-of-interest with discriminating information about the corresponding patients. It would be interesting to see an experimental analysis on this issue.

The proposed methodology is general and does not make any assumption on the size of the probes – neither on dataset size (as in numbers of WSIs), nor in the sizes of individual WSIs. The experiment (cE-2b) already hints in this direction, exploring the amount of overlap of the image data needed for re-identification. There can be other questions asked, such as when releasing data as a patches, how small patch still uniquely identify a WSI (for matching very small probes to the WSIs), but we intentionally did not include this in this work as there are many other questions that can be explored for specific cases and we expect this to be explored in the future. For effective search of patches from one (identical) WSI, one can consider using PatchTable [6] and this can be further explored as one of the specific questions in future work.

Reviewer #3

- Section 2.3.2, Setting E-2a: are these 158 pairs taken from one patient or different ones?

From different patients – we have updated the paper accordingly. Note there has been also a typo fixed, the number should have been 151, which has also been corrected.

Reviewer #3

- Section 2.3.2: It is unfortunate that settings H-4 and H-5 are not evaluated. In this reviewer's opinion they could yield interesting results.

This is intended for the international competition to be organized on this topic, as proposed in the paper.

Reviewer #3

- Table 1: Please explain what the 'autoencoders' column indicates. Is it another feature extraction approach tested but not described in the main part of the paper?

We have removed Table 1 from the manuscript (substituted with the box plot and graph figures) and we have also removed all autoencoder-based results completely. The autoencoder results had been added to the table in our broader investigation of the performance of different methods, but since autoencoders performed poorly they were not adding any value to the paper, in our opinion.

Reviewer #3

- Figure 8: Why is the y-axis label named 'score'? Shouldn't it be V_{att} ?

All the graphs have been revised and axes labelled properly. Based on requests of the other reviewers, the tables have been converted to additional graphs to better characterize distributions.

Reviewer #3

- Table 4: Please indicate the actual overlap (in percentage) of the WSIs to get a better understanding of the attack and its results.

Done.

Reviewer #3

Overall, while the paper is well-written and easy to follow there exist some typos that should be fixed. Some examples:

Typos:

Section 1: “[...] known as digital pathology.[2]”

Section 1: “[...] in the diagnostic process.[3]”

Section 1: “[...] digitized using an slide scanner [...]”

Section 1: “[...] but does necessarily appear in the dataset [...]”

Section 2.3.1: “[...] WSI probes to a patients' WSIs [...]”

Section 4: “[...] caution needs to exercised to mitigate [...]”

Paper has been proofread by a native speaker again and detected problems have been fixed.

Reviewer #3

References:

- [1] G. Smith, *On the foundations of quantitative information flow*. In: *International Conference on Foundations of Software Science and Computational Structures*. Springer, Berlin, Heidelberg, 2009. S. 288-302.
- [2] M. Alvim, et al. *The Science of Quantitative Information Flow*. Springer International Publishing, 2020.

Thank you for suggesting these materials! We have used them to provide a simple mapping of our extended model to the QIF.

References

1. Papernot, N. et al. *The limitations of deep learning in adversarial settings* in *2016 IEEE European symposium on security and privacy (EuroS&P)* (2016), 372–387.
2. El Emam, K. Risk-based de-identification of health data. *IEEE Security & Privacy* **8**, 64–67 (2010).
3. El Emam, K. *Guide to the de-identification of personal health information* (CRC Press, 2013).
4. Lomné, V., Prouff, E., Rivain, M., Roche, T. & Thillard, A. *How to estimate the success rate of higher-order side-channel attacks* in *International Workshop on Cryptographic Hardware and Embedded Systems* (2014), 35–54.
5. Hirano, H., Minagi, A. & Takemoto, K. Universal adversarial attacks on deep neural networks for medical image classification. *BMC medical imaging* **21**, 1–13 (2021).
6. Barnes, C., Zhang, F.-L., Lou, L., Wu, X. & Hu, S.-M. Patchtable: Efficient patch queries for large datasets and applications. *ACM Transactions on Graphics (ToG)* **34**, 1–10 (2015).

REVIEWER COMMENTS

Reviewer #3 (Remarks to the Author):

Summary:

The paper performs a privacy risk analysis of sharing whole-slide image (WSI) data widely used in digital pathology for diagnosis and machine learning model training. In particular, the authors develop a model that estimates the privacy risk via identity disclosure attacks, i.e., by computing how many patients in a WSI dataset can be successfully identified (after this has been shared) by an adversary with access to some partial knowledge about it. The proposed model is implemented following a methodology based on feature extraction via deep learning and similarity matching. Using real-world WSI data the authors experimentally evaluate how factors such as spatio-temporal proximity and staining methods affect the privacy risk and then use their results to develop guidelines for WSI data sharing with low privacy risks.

Comments:

The paper is interesting and timely as WSI data sharing is crucial in digital pathology for developing better insights about diseases and building (machine learning) tools that can assist medical doctors in their diagnosis. To the best of this reviewer's knowledge, this is the first paper that analyzes the privacy risks associated with WSI data sharing and can be a milestone that will likely spur further research on the topic. Moreover, the paper's methodology is reasonable and serves well the purpose of demonstrating the privacy risks of WSIs in an easy-to-understand manner that even non-expert audience would grasp. Other strong points of the paper include the usage of real-world WSI data for the experimental evaluation, the extensive analysis of factors (e.g., spatio-temporal proximity and staining methods) that affect the patient identification, as well as the proposal of guidelines for minimizing privacy risks when sharing WSI data. Nonetheless, this reviewer has a few concerns which are discussed below.

In particular, this reviewer is concerned about the estimation of the privacy risk by the proposed model: The computed risk is conditional to the presence of a patient data both in the attacker's background knowledge and the patient dataset to be shared. First, this assumption might be deemed as unrealistic depending on the setting and the authors should describe in more clarity the envisioned attackers, i.e., who they might be, how they might have access to this side knowledge (and how this is linked to the WSI hierarchy described), and what would be their motivation for launching the attack. While this reviewer understands that the authors consider such an attacker to estimate sufficiently the privacy risk, a crucial second question is whether this attacker is the "worst-case" attacker or not and accordingly if their method yields some bound on privacy loss or possibly under-estimates it. For instance, the model defined for the attack is not based on some known privacy definition, e.g., differential privacy, k-anonymity or similar. Moreover, the authors leave the additional information that might be inferred from a WSI fingerprint as an open question: This reinforces the above concerns about under-estimating the privacy risk and not relying on formal definitions (e.g., differential privacy) that can possibly bound such arbitrary leakage. Finally, the privacy risk considered is deterministic as it captures the number of patients uniquely re-identified (i.e., with probability 1) in the shared dataset. The authors should also take into account probabilistic adversaries (e.g., see Bayes vulnerability [1, 2]) which are likely to yield additional interesting results in terms of privacy risk.

While the methodology employed to instantiate the privacy model is simple and clear the paper does not reason about nor analyze sufficiently its effectiveness. For instance, it is unclear why the model is instantiated with two similarity metrics (cosine vs. euclidean) for the matching purpose: the paper's results follow a similar trend with both metrics so a reader is left wondering why both metrics are important or which one to choose for future evaluations. Similarly, there is no explanation why five different feature extractors are used: Is one approach better than the rest and why is that the case? While there does not seem to be a connection between the privacy risk estimation and the number of features extracted (k) the authors should analyze what qualitative properties of the deep learning

models used are important for realizing the proposed framework and why would they be suitable for WSIs in particular. Overall, the methodology could benefit from such a rigorous analysis to convince the reader about the experimental choices made.

The last part of the paper proposing guidelines for low-risk sharing of WSI data is interesting and important. However, a reader would expect to find an explicit connection between the proposed guidelines and the experimental results of the paper along with the threat model assumptions that justify the result. Moreover, the authors define an epsilon-safe dataset as part of their model definition, but this is nowhere used nor discussed in these guidelines. Could this privacy definition be used in practice as a technical tool to release a dataset with low privacy risk? If yes, how would someone tune the epsilon parameter and if not, why so? Additionally, it is unclear if the proposed privacy model is accompanied by an open-source tool that practitioners could make use of to aid in their decisions of sharing WSI data? Finally, the guidelines proposed by the authors seem to be missing an important component: Where is the patient consent involved in the data sharing process? Should not the patients be informed about the privacy risks estimated and make a decision about their WSI data being shared?

More detailed comments:

- Section 2.3: In the proposed methodology it seems that the probe needs to be of the same size as the WSI in the shared dataset. This reviewer is wondering if, e.g., substantially smaller probes (in terms of # of pixels) could be sufficient to identify a patient as long as they are derived from a point-of-interest with discriminating information about the corresponding patients. It would be interesting to see an experimental analysis on this issue.
- Section 2.3.2, Setting E-2a: are these 158 pairs taken from one patient or different ones?
- Section 2.3.2: It is unfortunate that settings H-4 and H-5 are not evaluated. In this reviewer's opinion they could yield interesting results.
- Table 1: Please explain what the 'autoencoders' column indicates. Is it another feature extraction approach tested but not described in the main part of the paper?
- Figure 8: Why is the y-axis label named 'score'? Shouldn't it be V_{att} ?
- Table 4: Please indicate the actual overlap (in percentage) of the WSIs to get a better understanding of the attack and its results.

Overall, while the paper is well-written and easy to follow there exist some typos that should be fixed. Some examples:

Typos:

Section 1: "[...] known as digital pathology.[2]"

Section 1: "[...] in the diagnostic process.[3]"

Section 1: "[...] digitized using an slide scanner [...]"

Section 1: "[...] but does necessarily appear in the dataset [...]"

Section 2.3.1: "[...] WSI probes to a patients' WSIs [...]"

Section 4: "[...] caution needs to exercised to mitigate [...]"

References:

[1] G. Smith, On the foundations of quantitative information flow. In: International Conference on Foundations of Software Science and Computational Structures. Springer, Berlin, Heidelberg, 2009. S. 288-302.

[2] M. Alvim, et al. The Science of Quantitative Information Flow. Springer International Publishing, 2020.

Comments after the revision:

This reviewer would like to thank the authors for revising the paper and acknowledges that it now addresses the reviewers' main concerns and has significantly improved. In particular, the authors have clarified the adversarial model by introducing the prosecutor model and discussed the possible extension of the model to probabilistic attacks. In terms of experiment method, the paper now employs a single similarity measure (cosine) and the experimental results are presented with box-plots (demonstrating the min/max/median of results). Finally, the paper's conclusions have been more tightly connected with the experimental results of Section 3 and user/patient consent has been discussed.

Two issues that were not addressed in this revision are the following:

First, the authors discuss probabilistic attacks only as an extension and they do not analyze their performance using the argument that such results might be hard to grasp by non-specialists. While this reviewer understands this, this is a scientific paper that can definitely include such an analysis.

Second, the rationale behind using multiple feature extractors for the experiments was not analyzed. It seems that there is a clear winner (SimCLR) among the deep learning models used but the authors decide to present results for all of them without investigating the reasons behind the performance for each of them.

Without this analysis it is hard to appreciate the value of going through all these results (and plots).

Nonetheless, the above two issues are not show-stoppers for the paper given its other strengths and it can be published by including a few discussions around these concerns.

Some remaining typos:

Section 1, page 5: "[.] then be able enrich [..]"

Figure 3, page 7: The set of probes should be \mathcal{P} and not \mathcal{I}

Section 2.1, page 8: "[.] any types randomized algorithms."

Section 2.2, page 8: "[.] can make two WSI [..]"

Section 2.3.3, page 14: "We studied influence [..]"

Section 3.1, page 17: "The resulting R_s statistics is [..]"

Section 4.1.1, page 24: "[.] but also the their [..]"

Reviewer #4 (Remarks to the Author):

The authors have addressed most of R1's previous comments. There are some remaining points need to be further clarified.

1) R1 pointed out that experimental evaluations were not including WSI hierarchy. The authors did not add experiments with some justifications. In this case, I suggest the authors consider to weaken the claims in subsection 2.2 since this part is not experimentally validated nor theoretically guaranteed.

2) The previous comments of "Finally, the deep learning models used as feature extractors are coming from general-purpose and trained models with natural images for computer vision tasks. It is needed experiments using specific experiments using publicly available deep learning method trained or fine-tuned with histopathology images." is not fully answered. The authors should present exact numbers as observed in the experiments which were claimed done.

Reviewer #5 (Remarks to the Author):

I was asked to evaluate the responses to Reviewer #2 in this manuscript. In my opinion, the authors have clearly and concisely responded to every point which was raised. This is an interesting study which seems technically sound and presents an interesting approach to an understudied problem.

Note that reviewers #1 and #2 have not responded to the editor and hence are not included in this round of responses.

Response to Reviewer #3

Reviewer #3

This reviewer would like to thank the authors for revising the paper and acknowledges that it now addresses the reviewers' main concerns and has significantly improved. In particular, the authors have clarified the adversarial model by introducing the prosecutor model and discussed the possible extension of the model to probabilistic attacks. In terms of experiment method, the paper now employs a single similarity measure (cosine) and the experimental results are presented with box-plots (demonstrating the min/max/median of results). Finally, the paper's conclusions have been more tightly connected with the experimental results of Section 3 and user/patient consent has been discussed.

We would like to thank to the reviewer for providing valuable feedback again!

Reviewer #3

*Two issues that were not addressed in this revision are the following:
First, the authors discuss probabilistic attacks only as an extension and they do not analyze their performance using the argument that such results might be hard to grasp by non-specialists. While this reviewer understands this, this is a scientific paper that can definitely include such an analysis.
Second, the rationale behind using multiple feature extractors for the experiments was not analyzed. It seems that there is a clear winner (SimCLR) among the deep learning models used but the authors decide to present results for all of them without investigating the reasons behind the performance for each of them. Without this analysis it is hard to appreciate the value of going through all these results (and plots).
Nonetheless, the above two issues are not show-stoppers for the paper given its other strengths and it can be published by including a few discussions around these concerns.*

For the first issue, we considered various extensions to the experiments to also test randomized attacks (e.g., attacking by distributions, or using various levels of thresholds), but all of them would excessively extend the length of the section text due to at least one more dimension being added to the existing schema of experiments. Hence, we propose to leave this aspect to a follow-up paper.

Regarding the second issue, the purpose of using different feature extractors is to demonstrate their variability – the text has been updated to clearly indicate this (Section 2.3.1, paragraph “Feature extraction.”). Given the request of Reviewer #4, a VGG16 model trained

specifically for prostate cancer detection has been added, though it shows inferior performance compared to other feature extraction strategies – as indicated in the previous response to the reviewers. Again, we believe an in-depth analysis of the feature extractor behavior would be a good target for a follow-up paper and for the international competition we are planning to organize.

We hope that, as you suggested in your review, these issues are not show-stoppers and we can proceed with the publication.

Reviewer #3

Some remaining typos:

Section 1, page 5: “[..] then be able enrich [..]”

Figure 3, page 7: The set of probes should be \mathcal{D} and not I

Section 2.1, page 8: “[..] any types randomized algorithms.”

Section 2.2, page 8: “[..] can make two WSI [..]”

Section 2.3.3, page 14: “We studied influence [..]”

Section 3.1, page 17: “The resulting R_s statistics is [..]”

Section 4.1.1, page 24: “[..] but also the their [..]”

Typos have been corrected.

Response to Reviewer #4

Reviewer #4

The authors have addressed most of R1’s previous comments. There are some remaining points need to be further clarified.

We would like to thank the reviewer for stepping in and providing valuable feedback!

Reviewer #4

1) R1 pointed out that experimental evaluations were not including WSI hierarchy. The authors did not add experiments with some justifications. In this case, I suggest the authors consider to weaken the claims in subsection 2.2 since this part is not experimentally validated nor theoretically guaranteed.

We believe this namely referred to paragraph on Distinguishability in the Section 2.2. The corresponding claims have been weakened and it has been explicitly made clear we have only tested (H-1) and (H-2).

Reviewer #4

2) The previous comments of “Finally, the deep learning models used as feature extractors are coming from general-purpose and trained models with natural images for computer vision tasks. It is needed experiments using specific experiments using publicly available deep learning methods trained or fine-tuned with histopathology images.” is not fully answered. The authors should present exact numbers as observed in the experiments which were claimed done.

We have added VGG16 feature extractor denoted as VGG16histo, which has been trained for prostate cancer diagnosis and delivers state-of-the-art performance. As expected this model indeed delivers inferior performance as it has been trained for detecting features from detailed tissue structures on individual tiles extracted from whole-slide images.

Response to Reviewer #5

Reviewer #5

I was asked to evaluate the responses to Reviewer #2 in this manuscript. In my opinion, the authors have clearly and concisely responded to every point which was raised. This is an interesting study which seems technically sound and presents an interesting approach to an understudied problem.

We would like to thank the reviewer for stepping in and providing feedback!

REVIEWERS' COMMENTS

Reviewer #4 (Remarks to the Author):

The authors have addressed all the previous questions. I have no further comments.